# Heating up decision boundaries: isocapacitory saturation, adversarial scenarios and generalization bounds

**Bogdan Georgiev**
Fraunhofer IAIS, ML2R
bogdan.m.georgiev@gmail.com

**Lukas Franken**
Fraunhofer IAIS, ML2R, University of Cologne
lukas.b.franken@gmail.com

**Mayukh Mukherjee**
IIT Bombay
mathmukherjee@gmail.com

## Abstract

In the present work we study classifiers' decision boundaries via Brownian motion processes in ambient data space and associated probabilistic techniques. Intuitively, our ideas correspond to placing a heat source at the decision boundary and observing how effectively the sample points warm up. We are largely motivated by the search for a soft measure that sheds further light on the decision boundary's geometry. En route, we bridge aspects of potential theory and geometric analysis (Maz'ya (2011); Grigor'Yan & Saloff-Coste (2002)) with active fields of ML research such as adversarial examples and generalization bounds. First, we focus on the geometric behavior of decision boundaries in the light of adversarial attack/defense mechanisms. Experimentally, we observe a certain capacitory trend over different adversarial defense strategies: decision boundaries locally become flatter as measured by isoperimetric inequalities (Ford et al. (2019)); however, our more sensitive heat-diffusion metrics extend this analysis and further reveal that some non-trivial geometry invisible to plain distance-based methods is still preserved. Intuitively, we provide evidence that the decision boundaries nevertheless retain many persistent "wiggly and fuzzy" regions on a finer scale.

Second, we show how Brownian hitting probabilities translate to soft generalization bounds which are in turn connected to compression and noise stability (Arora et al. (2018)), and these bounds are significantly stronger if the decision boundary has controlled geometric features.

## 1 Introduction and background

The endeavor to understand certain geometric aspects of decision problems has lead to intense research in statistical learning. These range from the study of data manifolds, through landscapes of loss functions to the delicate analysis of a classifier's decision boundary. In the present work we focus on the latter. So far, a wealth of studies has analyzed the geometry of decision boundaries of deep neural networks (DNN), reaching profound implications in the fields of adversarial machine learning (adversarial examples), robustness, margin analysis and generalization. Inspired by recent isoperimetric results and curvature estimates (Ford et al. (2019); Moosavi-Dezfooli et al. (2019); Fawzi et al. (2016)), we attempt to provide some new aspects of decision boundary analysis by introducing and studying a corresponding diffusion-inspired approach.

In this note the guiding idea is to place a heat source at the classifier's decision boundary and estimate its size/shape in terms of the amount of heat the boundary is able to emit within a given time (Fig. 1). The goal is to extract geometric information from the behavior of heat transmission. This technique of **heat content** seems well-known within capacity/potential theory and has led to a variety of results in spectral analysis relating heat diffusion and geometry, Jorgenson & Lang (2001); Grigor'Yan & Saloff-Coste (2002); Maz'ya (2011). However, working with such heat diffusion directly in

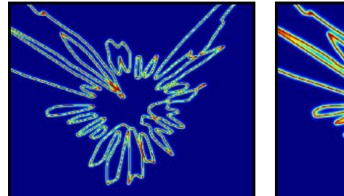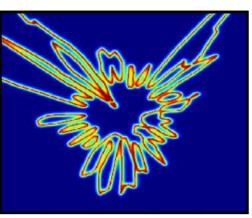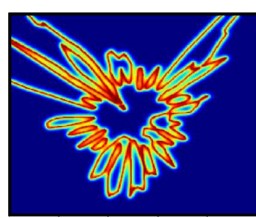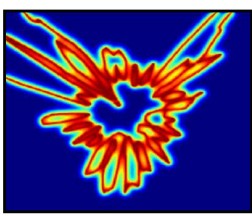

Figure 1: Heating up a planar decision boundary of a 5-layer MLP over time. The amounts of radiated heat reflect the geometry of the decision boundary: size, density, curvature.

terms of the corresponding differential equations is impractical. To this end, we note that, due to Feynman-Kac duality, the heat estimates are convertible to Brownian motion hitting probabilities. Thus we circumvent the need for solving intractable differential equations and instead are able to employ a straightforward Monte-Carlo sampling scheme in the ambient data space (Section 3).

**Background on defense training** We apply the above analysis in the context of adversarial machine learning (Section 4) where one studies the interaction between an adversary and a ML system. One of the goals of the subject is to design attack/defense training strategies improving the robustness of a given ML model - in the present work we are interested in how adversarial/noise defense training are reflected geometrically. Many different metrics to estimate robustness have been proposed: on one hand, there is *adversarial robustness* (the probability that error samples lie very near a given data point $x$); on the other hand, there is *corruption robustness* (the probability of getting an error sample after perturbing a given data point $x$ with some specified noise). In our context, heat diffusion naturally suggests a **capacitory robustness** metric: this metric is built upon the probability that Brownian motion started at a given data point $x$ will hit error samples within a given time window. One can perceive this metric as a *combination of adversarial and noise robustness* (Brownian motion has continuous paths and specified stopping time determined by boundary impact). In this perspective, our work is aligned with studies of other robustness metrics and curvature results (cf. Fawzi et al. (2016) for a "semi-random" projection robustness and relations to curvature). We study the capacitory metric on the well-known CIFAR10 and MNIST datasets and observe that defense training techniques may either yield a certain (although not substantial) decrease (noise training) or fail to have a significant effect on continuous Brownian attacks overall. Surprisingly, in both cases the studied capacitory metric does not converge to the corresponding value as in the case of a flat decision boundary. Due to our comparison statements and curvature considerations, this means that locally around clean data points the geometry is in general flattened out but may still retain complexity and substantial areas of (small) non-vanishing curvature. In other words, from the point of view of our heat diffusion metrics, decision boundaries locally exhibit non-flat behaviour.

**Background on generalization estimates** Finally, we observe that the collected heat/hitting-probability metrics can further be used to obtain generalization bounds where, in a nutshell, one evaluates the performance of a model on unseen data in terms of the performance over a given sampled data, the model's expressiveness, dimension, etc. In this regard, we view decision boundary heat diffusion traits as an indicator of how noise-stable a given model is - this relates Brownian hitting bounds with recent compression-based generalization techniques in the spirit of Arora et al. (2018); Suzuki et al. (2018; 2020). More precisely, we proceed in two steps: first, we construct a "smaller" compressed model that is almost equivalent to the initial one in an appropriate heat-theoretic way; second, we obtain generalization estimates for the smaller model in terms of the decision boundary hitting probabilities (computed on the empirical dataset). Furthermore, the bounds are significantly improved under additional geometric assumptions on the decision boundary of the initial model.

**Additional related work** The interplay between *heat diffusion and geometry* lies at the heart of many topics in geometric analysis and spectral theory (cf. Jorgenson & Lang (2001); Grigor'Yan (2001) for a far reaching overview). Some direct applications of heat diffusion techniques to zero sets of eigenfunctions are seen, for example, in Steinerberger (2014); Georgiev & Mukherjee (2018a;b). The literature on *adversarial ML* is vast: to name a few central works in the field, we refer to Dalvi et al. (2004); Biggio & Roli (2018); Szegedy et al. (2014). Much effort has been invested in designing

and understanding strategies that will render a model robust to various attacks (e.g. Madry et al. (2018); Carlini & Wagner (2017)). In particular, the geometry of decision boundaries has been the focus of many works in the subject leading to breakthroughs in curvature estimates, boundary flatness and robustness, schemes for detecting boundary complexity, proposing adversarial attacks/defenses and diffusion based techniques towards constructing decision boundary from partially pre-labelled data (e.g. Ford et al. (2019); Fawzi et al. (2016; 2017; 2018); Dezfooli et al. (2018); Moosavi-Dezfooli et al. (2019); Karimi et al. (2019); Karimi & Tang (2020); He et al. (2018); Szlam et al. (2008)). *The theory of generalization bounds* has formed a classical main line of ML and statistical inference research (Vapnik (1999)). In this direction central questions address the generalization properties of heavily over-parametrized deep neural network models. According to some classical VC-dimension results such models should overfit the data and generalize poorly. Extensive research effort has been invested in developing appropriate sharper techniques to explain generalization of DNN models: on one hand there are the methods based on norm estimation whose bounds are not explicitly using the number of the network's parameters (see Golowich et al. (2019); Neyshabur et al. (2015; 2018); Wei & Ma (2019); Bartlett et al. (2017), etc). On the other hand, recent results based on compression and VC-dimension can lead to sharper bounds (Arora et al. (2018); Suzuki et al. (2018; 2020)).

## 2 CONTRIBUTIONS, CONTEXT AND PAPER OUTLINE

An outline of our essential contributions is given as follows:

1. We analyze decision boundary geometries in terms of novel heat diffusion and Brownian motion techniques with thorough theoretical estimates on curvature and flattening.

2. We show, both theoretically and empirically (in terms of adversarial scenarios on state-of-art DNN models), that the proposed heat diffusion metrics detect the curvature of the boundary; they complement, and in some respects are more sensitive in comparison to previous methods of boundary analysis - intuitively, our heat driven metrics are sharper on a finer scale and can detect small-scale "wiggles and pockets". As an application, we are thus able to provide evidence that adversarial defenses lead to overall flatter boundaries but, surprisingly, the heat traits do not converge to the corresponding flat-case, and hence, finer-scale non-linear characteristics (e.g. "wiggles and pockets") are persistent.

3. Moreover, the preservation of "wiggles and pockets" means that susceptibility to naive Brownian motion attacks is not significantly decreased via adversarial defense mechanisms.

4. Finally, we introduce a novel notion of compression based on heat diffusion and prove that stability of heat signature translates to compression properties and generalization capabilities.

In terms of context, the present note is well-aligned with works such as Ford et al. (2019); Dezfooli et al. (2018); Fawzi et al. (2016; 2018). Among other aspects, these works provide substantial analysis of the interplay between geometry/curvature and adversarial robustness/defenses - in particular, we use some of the these tools (e.g. isoperimetric saturation) as benchmarks and sanity checks. However, in contrast, in our work we provide a *non-equivalent technique* to address decision boundary geometry for which we provide an extensive theoretical and empirical evaluation with insights on the preservation of finer-scale traits. Intuitively, previous distance-based geometric methods could be considered as a "coarser lens", whereas the present heat-diffusion tools appear to be much more sensitive. As a large-scale example, Brownian particles emanating from a point are able to distinguish between a decision boundary which is a hyperplane at distance $d$ and a decision boundary which is a cylinder of radius $d$ wrapping around the point. Our notion of compression is inspired by Arora et al. (2018), and establishes a connection between the Johnson-Lindenstrauss dimension reduction algorithm with diffusion techniques. Furthermore, we bridge the proposed heat-theoretic techniques with generalization bounds in the spirit of Arora et al. (2018); Suzuki et al. (2020). In particular, this shows that overall lower heat quantities at sample points imply better generalization traits. A step-wise road map of the present work is given below:

- (Subsection 3.1) We start by discussing what heat diffusion is and how it is to be evaluated - here we discuss that, via Feynman-Kac duality, one can essentially work with Brownian motion hitting probabilities.

- (Subsections 3.2 and 3.3) We introduce **the isocapacitory saturation** $\tau$ - a heat-theoretic metric that will be used to estimate boundary flatness. Moreover, here we emphasize the

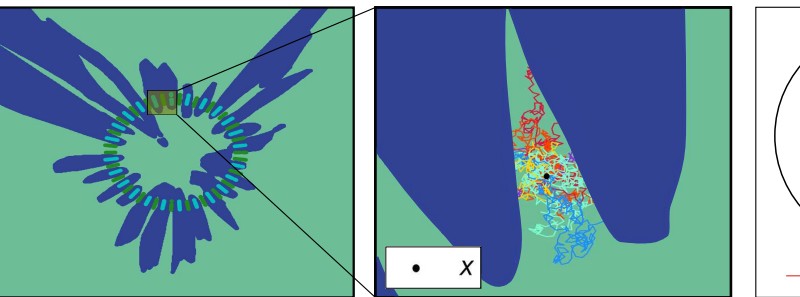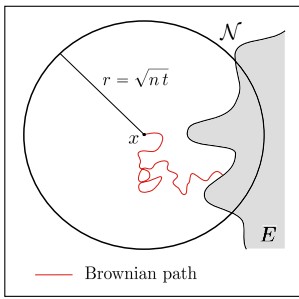

Figure 2: A planar 2-class dataset that alternates along a circle. *(Left)* A depiction of the planar circle-like dataset and the corresponding decision boundary of a 5-layer MLP. *(Center)* Brownian paths starting at a data point $x$ and killed upon impacting the decision boundary/opposite class. *(Right)* Set-up of the local Brownian motion analysis with notation on radius $r$, dimension $n$ and Brownian runtime $t$.

properties of $\tau$ such as relations to curvature (Proposition 3.1) and the *novel information* obtained from heat theoretic methods in comparison to previous distance-based ones.

- (Subsection 3.4) We compute $\tau$ for certain geometric model cases such as hyperplanes, cones, wedges and "spiky" sets (Lemmas 3.2 and 3.3). This allows us later to evaluate how much a given geometry resembles these model cases.

- (Section 4) Next, we are in a position to evaluate and compare $\tau$ for decision boundaries of DNNs. We experimentally illustrate the effect of adversarial defense mechanisms and noise robustness on $\tau$ (PGD/FGSM on MNIST and CIFAR-10).

- (Section 5) We prove that heat transmission relates to generalization bounds (Propositions 5.1 and 5.2) - in particular, lower levels of heat at sample points yield sharper generalization bounds. Finally, we complete the discussion by informally stating our compression scheme.

- (Appendix) Our methods leverage several tool sets extensively. For this reason our goal in the main text is to only collect and showcase the techniques and results. However, the thorough in-depth analysis is provided in the Appendix where the reader can find all relevant proofs and further background and references.

## 3 MOTIVATION AND MAIN IDEAS

### 3.1 GEOMETRY SEEN THROUGH BROWNIAN MOTION AND DIFFUSION

**Notation**    Let us consider a dataset $\mathcal{X} := \{(x_i, y_i)\}_{i=1}^m$ consisting of feature points $x_i \in \mathbb{R}^n$ and their corresponding labels $y \in \{1, \ldots, k\}$. Let us suppose that a $k$-label classifier $f : \mathbb{R}^n \to \mathbb{R}^k$ labels a point $x \in \mathcal{X}$ as $\arg\max_{i \in [1,k]} f(x)[i]$. The **decision boundary** of $f$ is given by $\mathcal{N} := \{x \in \mathbb{R}^n | f(x) \text{ has two or more equal coordinates}\}$ (cf. Fig. 2). Assuming $f$ is sufficiently regular, one thinks of $\mathcal{N}$ as a collection of hypersurfaces in $\mathbb{R}^n$. Further, for a given target label $y$ we define the **target (error) set** $E(y)$ as the set of points on which the classifier's decision is different from $y$, i.e. $E(y) := \{x \in \mathbb{R}^n | \arg\max_{i \in [1,k]} f(x)[i] \neq y\}$ (here we remark that if $\arg\max$ is set-valued at $x$ with several coordinates obtaining the maximum value, then by convention $x$ is contained in $E(y)$). Clearly, if a given data sample $(x_0, y_0) \in \mathcal{X}$ is correctly classified by $f$, then $x_0$ is outside of the error set $E(y_0)$. Finally, we note that the boundary of $E(y)$ coincides with $E(y) \cap \mathcal{N}$ and moreover, $\mathcal{N}$ is the union of the boundaries of $E(y)$ for all labels $y$.

**Feynman-Kac duality and hitting probabilities**    As mentioned in Section 1 we wish to study a heat diffusion process where we place a heat source at the decision boundary $\mathcal{N}$: formally, this is given by a heat equation with appropriate initial and boundary conditions (Appendix, Subsection A.2). Avoiding the impracticality of working with the differential equations directly, we bring forward the theorem of Feynman-Kac that relates the solution of the diffusion process to hitting probabilities of Brownian motion (Appendix, Subsection A.3). By way of notation, for an open set $U \subseteq \mathbb{R}^n$, let

$\psi_U(x,t)$ denote the probability that a Brownian particle starting at the point $x$ will enter $U$ within time $t$. In other words,

$$\psi_U(x,t) := \mathbb{P}_{\omega \sim \mathbb{W}}\left[\exists\, t_0 \in [0,t] \mid \omega(t_0) \in U\right], \quad x \in \mathcal{X}, \tag{1}$$

where $\omega$ denotes a Brownian motion defined over the interval $[0,t]$ that follows the standard Euclidean Wiener distribution. The amount of heat that a point $x$ receives from $\mathcal{N}$ within time $t$ is comparable to the **hitting probability** that a Brownian particle starting at $x$ will impact the boundary within time $t$ (cf. Fig. 2). Provided that $x$ is correctly classified this is equivalent to the probability of impacting the decision boundary. In general, we evaluate $\psi_{E(y)}(x,t)$ (which we often denote by $\psi(x,t)$ by minor abuse of notation) through direct sampling; however, in some model cases, e.g. $E(y)$ being a half-space, a spherical shell or a conical set, $\psi(x,t)$ has a concise closed form (Subsection 3.4 below) that can be evaluated analytically. This allows us to easily measure deviations and compare the heat imprint of $\mathcal{N}$ to particular model cases.

**Local analysis and set-up**   As mentioned above our analysis is local. For each clean data point $x$ we consider a ball $B(x,r)$ centered at $x$ with radius $r$ and perform all our computations there. In particular, a free Brownian motion starting at $x$ and defined over a maximal time interval $[0,t]$ will on average travel a distance of $\sqrt{nt}$ (Appendix, Subsection A.1). This suggests to couple $r$ and the maximal Brownian running time $t$ via $r = \sqrt{nt}$ (cf. Fig. 2), so that, if not stopped by boundary impact, Brownian motion will, on average, reach the sphere $\partial B(x,r)$ by its maximal stopping time.

## 3.2   AN ISOPERIMETRIC AND ISOCAPACITORY PERSPECTIVE

**Isoperimetric results**   Isoperimetric estimates will be the starting baseline (Ford et al. (2019)) to detect low levels of curvature and boundary flatness. For some background in isoperimetric results we refer to (Appendix, Subsection A.4). Let us start by defining the **relative error volume**

$$\mu(x,r) := \frac{\text{Vol}(E(y) \cap B(x,r))}{\text{Vol}(B(x,r))}. \tag{2}$$

We recall the so-called *Gaussian isoperimetric inequality* Borell (1975); Ford et al. (2019):

$$\tilde{d} \leq -\frac{r\,\Phi^{-1}(\mu)}{\sqrt{n}}, \quad \mu \leq 1/2, \tag{3}$$

where $\Phi^{-1}$ denotes the inverse standard normal c.d.f. and where $\tilde{d} = d(\tilde{x}, \mathcal{N}_f)$ denotes the median distance with $\tilde{x}$ varying normally and concentrated in the ball $B(x,r)$, and $\tilde{d} = 0$ if $\mu \geq 1/2$. Here the isoperimetric result is rigid in the sense that equality in (3) occurs only if $E(y)$ is a half-space. In Ford et al. (2019) the authors demonstrate that defense training mechanisms lead to decision boundaries that saturate this isoperimetric inequality, i.e. in this isoperimetric sense, the decision boundary $\mathcal{N}$ becomes locally closer to being a flat hyperplane. We define the ratio between the LHS and RHS in eq. (3) as the **isoperimetric saturation**.

**Isocapacitory results**   In our context of hitting probabilities (eq. (1)), results in potential theory allows us to prove **isocapacitory bounds** which are similar in spirit to isoperimetric bounds. More precisely one has:

$$\mu(x,r) \leq c_n\,\psi(x,t)^{\frac{n}{n-2}}, \tag{4}$$

where $c_n$ is an appropriate constant depending on the dimension $n$, and $r = \sqrt{nt}$. The proof relies on potential theory tools (capacity) and can be found in Appendix, Proposition A.3. Motivated by the above isoperimetric saturation results, one of our main goals is to study how $\mu$ compares to $\psi(x,t)$. To this end we define the **isocapacitory saturation** $\tau$ as

$$\tau(x,r) := \frac{\psi(x,t)^{\frac{n}{n-2}}}{\mu(x,r)}. \tag{5}$$

The basic guiding heuristic is that high values of $\tau$ indicate that $E(y)$ has a very low volume in comparison to its boundary size and respective heat emission. This is the case whenever $E(y)$ is a very thin region with a well-spread boundary of large surface area - e.g. a set that resembles thin spikes entering the ball $B(x,r)$. In contrast, lower values of $\tau$ should indicate a saturation of the isocapacitory inequality (4) and imply that $E(y)$ has a volume that is more comparable to its heat emission - e.g. thicker sets with tamer boundary. To quantify this intuition, we explicitly evaluate $\tau$ for some model scenarios (Subsection 3.4).

### 3.3 The novel information given by heat diffusion

**Distances vs. hitting probabilities**   As discussed above, several works investigate decision boundaries in terms of distance-based analysis (Ford et al. (2019); Fawzi et al. (2016); Karimi & Tang (2020); Karimi et al. (2019)). We remark that our analysis based on hitting probabilities augments and extends the mentioned distance-based approaches. Although related, the two concepts are not equivalent. A guiding example is given by $E(y)$ being a dense collection of "thin needles" (Appendix, Subsections A.4, A.5); in such a scenario the average distance to $\mathcal{N}$ is very small, as well as the chance a Brownian particle will hit $\mathcal{N}$. On the other hand, if $\mathcal{N}$ is a dense collection of hyperplanes, the average distance to $\mathcal{N}$ is again small, but Brownian motions almost surely will hit $\mathcal{N}$. In this sense, evaluating hitting probabilities yields a different perspective than is available from distance-based analysis and sheds further light on the size and shape of the decision boundary, particularly with regards to its capacity and curvature features.

**Isoperimetric vs. isocapacitory saturation**   Another demonstration of the additional information obtained through $\tau$ is given by almost flat shapes in higher dimensions that saturate isoperimetric bounds (Appendix, Subsection A.4). In these scenarios small geometric deformations can have a significant impact on $\tau$, and at the same time almost preserve isoperimetric bounds. In other words $\tau$ provides an additional level of geometric sensitivity. We discuss this further in Section 4.

**The effect of curvature**   The interplay between curvature of the decision boundary and robustness has been well studied recently, e.g. Fawzi et al. (2016); Moosavi-Dezfooli et al. (2019) where various forms of robustness (adversarial, semi-random and their ratio) have been estimated in terms of the decision boundary's curvature. Intuitively, the differential geometric notion of curvature measures how a certain shape is bent. The precise definition of curvature involves taking second-order derivatives which is in most cases impractical. However, in our context we show that the isocapacitory saturation $\tau$ implies certain curvature bounds. These statements exploit relations between curvature and volume and lead to pointwise and integral curvature bounds. As an illustration, we have:

**Proposition 3.1** (Informal). *Let $(x, y) \in \mathcal{X}$ be a data sample. Then, provided that the distance $d(x, \mathcal{N})$ is kept fixed, larger values of $\tau$ locally imply larger pointwise/integral curvature values.*

A deeper analysis with formal statements and additional details are provided in Appendix, Subsection A.6. The advantages that curvature yields for some types of compression schemes and generalization bounds is also intensely investigated in Appendix, Section B.

### 3.4 Model decision boundaries: hyperplanes, wedges, cones and "spiky" sets

Given a certain geometric shape, one is often faced with questions as to how flat or spherical the given geometry is. To this end, a central technique in geometric analysis is comparing to certain model cases - e.g. a sphere, plane, saddle, etc. After having introduced $\tau$ and its basic traits we now evaluate it for several model cases (flat hyperplanes, wedges, cones, balls and "spiky" sets). Each of these model cases illustrates a distinguished $\tau$-behaviour: from "tame" behaviour (hyperplanes, balls) to explosion (thin cylinders, "needles and spiky" sets). Hence, having comparisons to these model cases and given an decision boundary, one can, quantify how far away is the given surface from being one of the models. We start by discussing the flat linear case:

**Lemma 3.2.** *Let $(x, y)$ be a data sample and suppose that $E(y)$ forms a half-space at a distance $d$ from the given data point $x \in \mathbb{R}^n$. Then*

$$\tau(x, r) = 2 \, \Phi \left( -\frac{d}{\sqrt{t}} \right) \frac{\operatorname{Vol}(B(x, r))}{V_n(d, r)}, \tag{6}$$

*where $\Phi(s)$ is the c.d.f. for the standard normal distribution, and $V_n(d, r)$ is the volume of the smaller $n$-dimensional solid spherical cap cut-off at distance $d$ from the center of a ball of radius $r$.*

The computation uses standard reflection principle techniques. Figure 3 depicts an experimental discussion on Lemma 3.2. Another illuminating model is given by a "spiky" set - e.g. a thin cylinder, which is in some sense the other extreme. We have

**Lemma 3.3** (Appendix, Subsection A.5). *Suppose that $E(y)$ is a cylinder of height $h$ and radius $\rho$ that enters the ball $B(x, r)$. Then $\tau \nearrow \infty$ as $\rho \searrow 0$.*

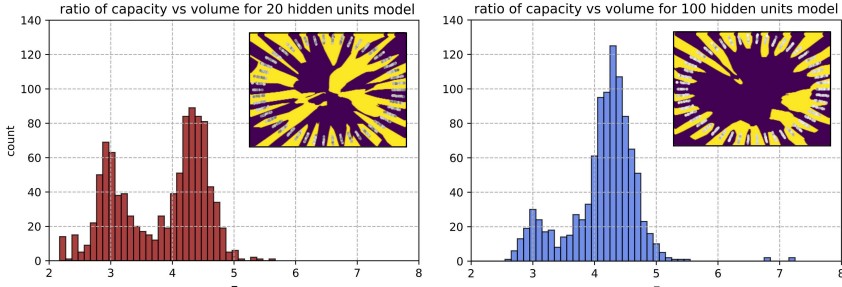

Figure 3: A visual depiction of decision boundaries and saturation $\tau$ for 5-layer MLP models with 20 and 100 hidden units trained over a planar "circular" dataset (depicted in grey). For each data sample $x$ the ball $B(x, r)$ is selected so that the relative volume $\mu(x, r)$ is 0.1. According to Lemma 3.2 a flat decision boundary would correspond to $\tau \approx 3.32$. *(Left)* The saturation $\tau$ exhibits a *bi-modal* behaviour with peaks around the values 3 and 4.3. These correspond to data points squeezed between thin elongated regions that locally closely resemble the flat case, or tinier "pockets" with higher curvature, respectively. *(Right)* The saturation $\tau$ is more closely concentrated around 4.3 and, accordingly, the decision boundary mainly consists of smaller "pockets" of higher curvature.

Further comparison results for additional model cases are given in Appendix, Subsection A.5.

## 4  ADVERSARIAL ATTACKS AND DEFENSES

**Background and set-up**   We now analyze how strategies for improving adversarial and noise shift robustness affect the decision boundary's heat diffusion properties. In particular, we keep track of Brownian hitting probabilities $\psi$ and the isocapacitory saturation $\tau$. On one hand, we can view $\psi$ as a **capacitory robustness** metric against continuous interpolation attacks given by Brownian noise (see also Section 1). On the other hand, Subsection 3.4 indicates how the behaviour of $\tau$ reveals deviation from the case of a flat or "spiky" and curvy decision boundary. Our *empirical analysis* uses the well-known CIFAR10 and MNIST datasets (details, preprocessing and enhancements are given in Appendix, Subsection C.5). For CIFAR10, we used the Wide-ResNet-28-10 (Zagoruyko & Komodakis (2016); Ford et al. (2019)) and ResNets with 32, 44 and 56 layers (He et al. (2016)). For MNIST, we selected a LeNet-5 and additional CNN architectures. Motivated by previous work (e.g. Ford et al. (2019)), we perform 3 types of training: ordinary stochastic gradient descent (ADAM optimization), training with Gaussian noise data augmentation and training with adversarial defense strategies (FGSM and PGD methods, see also Appendix, Section C.4 for details and remarks on robustness). Detailed outline of the numerics behind Brownian motion sampling, isoperimetric/isocapacitory saturation and relative volume sampling are given in Appendix, Subsection C.3.

**Analysis of results**   Recent results (Ford et al. (2019); Schmidt et al. (2017)) have shown qualitative differences between the adversarially robust boundaries of MNIST and CIFAR-10, which also impact the experimental findings in this work. In short, a robust decision boundary is in the MNIST case less spiky in comparison to CIFAR. For more details we refer to Appendix, Subsection C.2. In Fig. 4 we collect the statistics of the WRN and LeNet models on CIFAR10 and MNIST, respectively. On one hand, we confirm previous results (Ford et al. (2019); Fawzi et al. (2016)) implying the "flattening-of-boundary" phenomenon: noisy and adversarial training appear to improve and saturate isoperimetric bounds. Furthermore, the ball $B(x, r)$ realizing relative error volume $\mu$ of 1% is on average scaled up for adversarial and, especially, noisy training. On the other hand, an intriguing behaviour is observed for the decision boundary's heat diffusion traits. The isocapacitory saturation $\tau$ does not appear to concentrate around the value corresponding to a flat hyperplane: defense training strategies, both FGSM and PGD-based, may not have a significant impact on the behaviour of $\tau$ by forcing it to converge to the case of a flat decision boundary (shown as horizontal red punctured line). Put differently, the chance that a continuous Brownian perturbation will find an adversarial example (scaled to the appropriate ball $B(x, r)$) will not be significantly altered on average (see Appendix, Subsection C.7 for a visual reference). However, it appears that noisy training consistently delivers lower values of $\tau$ - intuitively, this is expected as the decision boundary is adjusted in terms

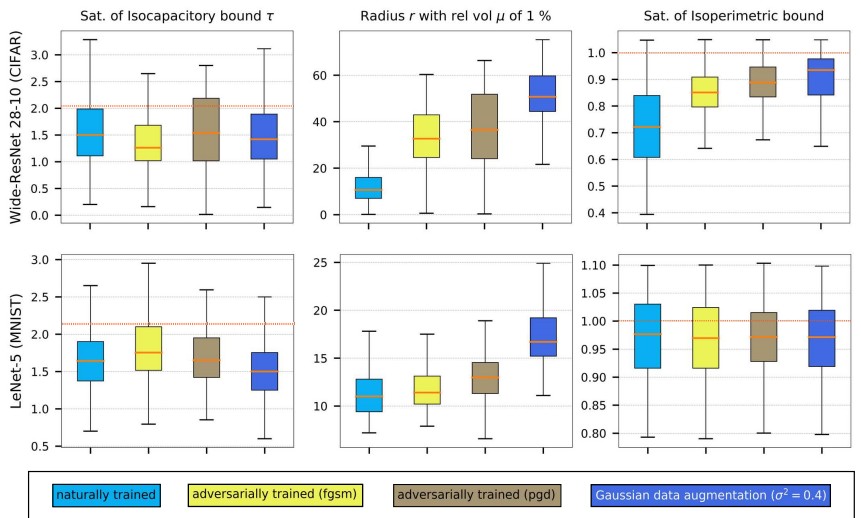

Figure 4: Results for a Wide-ResNet 28-10 and a LeNet-5 trained on CIFAR10 and MNIST, respectively. Different boxplots correspond to different training strategies: ordinary, adversarial, with noise or with a Brownian augmentation. Data is collected over 1000 test data points, where each radius $r$ is selected so that the relative error volume $\mu$ equals 1%. Left-to-right the columns correspond to the isocapacitory saturation $\tau$, the radius $r$ realizing $\mu = 1\%$ and the isoperimetric saturation. Finally, red punctured horizontal lines indicate the corresponding values for flat decision boundaries.

of adding Gaussian "blobs", thus naturally being rounder. Geometrically, the sensitivity of $\tau$ to small perturbations in almost flat surfaces (Subsection 3.2) indicates that locally around clean (unperturbed) data points an amount of curvature and more complex geometry are still retained. Of course, this amount is not as large as to violate saturation of isoperimetric bounds and robustness comparability results in the sense of Fawzi et al. (2016). For example, in the case of CIFAR10 a simple geometric model surface that has a similar $\tau$-behaviour (as for the adversarial and noisy training) is given in (Appendix, Subsections A.4, A.5): considering a data point $x$, an almost flat decision boundary that is concavely bent w.r.t. $x$ with approximate curvature of $\approx 1/(12.3r)$. These observations reveal finer properties concerning decision boundary flattening due to defense training: in particular, noisy training appears to flatten decision boundaries and slightly bend them concavely w.r.t. to the clean data points. Further results for ResNet models and CNN are provided in (Appendix, Subsection C.7).

**Spiky sets and control on** $\tau$    In Fig. 4 large outlying values of $\tau$ are filtered out. However, values of $\tau$ larger than 10 can occupy up to 1.3% for ordinary training and 2.1%, 2.6% for adversarial, noisy training, respectively. It follows, that the geometry of high-dimensional decision boundaries does not admit too many high-curvature (see also Proposition 3.1) spiky regions of low volume and high heat emission (high surface area) in the sense of Subsections 3.2, 3.4. However, it appears that defense training can increase the number of such spiky regions: one might explain such behaviour by seeing defense training as a bundle of additional geometric conditions that sometimes are not able to agree and thus lead to a more degenerate (singular) geometry. Further, with respect to the initial analysis of Fig. 4, a natural question is whether one can control $\tau$ along with the isoperimetric saturation - ultimately, one hopes to design better decision boundaries (flatter, or appropriately curved Moosavi-Dezfooli et al. (2019)) eventually leading to more robustness. However, getting a tight control on $\tau$ could be a difficult task. It is, indeed, possible to obtain some basic grip on $\tau$: we trained a LeNet-5 architecture on MNIST that exhibited significantly increased $\tau$ values and preserved isoperimetric saturation (statistics are shown as the rightmost boxplot in Fig. 4). Similar to many adversarial defenses, the training consisted in augmenting the dataset with attacks given in this case by Brownian paths. However, it seems difficult to force $\tau$ to concentrate around the flat-case value, as well as to obtain competitive robustness of the model. On one hand, this is explained via the need to control heat diffusion through Brownian motion - the mentioned naive method is not able to capture the hitting properties sufficiently well; on the other hand, as discussed above heat diffusion properties can be far more sensitive than isoperimetric saturation w.r.t. minor geometric perturbations.

## 5 Generalization bounds in terms of hitting probabilities

**Compression, noise stability and generalization** Recent advances (Arora et al. (2018); Suzuki et al. (2018; 2020)) indicate that generalization can be related to compression and noise stability. The guiding strategy is: (1) a large DNN $f$ that is stable against (layer-wise) noise injections admits an effective compression to a simpler model $\tilde{f}$ which is almost equivalent to $f$. Intuitively, the noise stability absorbs the defects introduced by compression; (2) concentration results imply generalization bounds for $\tilde{f}$. Admittedly, the generalization estimate is obtained initially for the smaller model; however, it is also possible to "transfer" the bound to $f$ (see the discussion at the end of this Section).

In this context a simple observation is that Brownian motion and its hitting probabilities can be related, respectively, to noise injection and margins of classification: small hitting probability of the decision boundary should indicate "margin-safety" and allow to compress parameters of the model more aggressively. However, in contrast to injecting normal noise, Brownian motion, with stopping time given by boundary impacts, is more delicate and requires further analysis of the decision boundary. In the following we propose a theoretical framework that, we hope, will augment and produce further insights into the interplay between noise stability and generalization bounds. The statements are inspired by the results in Arora et al. (2018); Suzuki et al. (2020) and we follow the notation therein. First, we propose several options for goodness of approximation (compression) in the sense of heat diffusion (Appendix, Subsection B.1). We give the following definition:

**Definition 1.** *Given a positive real number $\eta$, a classifier $g$ is said to be an $\eta-$compression of $f$ if*

$$\left| \psi_{E_g(y)}(x, \gamma^2) - \psi_{E_f(y)}(x, \gamma^2) \right| < \eta \tag{7}$$

*for all points $x$ in the training sample, labels $y$ and real numbers $\gamma$.*

Now, as mentioned above we have the following generalization bounds for the compressed model:

**Proposition 5.1.** *Let us suppose that $f$ is approximable by $g$ in the sense of Definition 1. Here $g \in A$, where $A$ is a family of classifiers $\mathbb{R}^n \to \mathbb{R}$ parametrized by $q$ parameters assuming $r$ discrete values. For a classifier $h$, let $C_h(x, y, t)$ be the event that a Brownian path starting at $x$ hits $E_h(y)$ within time $t$. Then for $t_1 \leq t_2 \leq T$ we have*

$$L_0(g) \leq \mathbb{P}_{(x,y)\sim D}\left(C_{g_\alpha}(x, y, t_1)\right) \leq \mathbb{P}_{(x,y)\sim \mathcal{X}}\left(C_f(x, y, t_2)\right) + \eta + O\left(\sqrt{\frac{q \log r}{m}}\right) \tag{8}$$

*with probability at least $1 - e^{-q \log r}$ and $L_0$ denoting the expected loss over the true data distribution.*

Taking $t_2 \to 0$ in (8), one recovers the empirical loss $\hat{L}_0(f)$ on the RHS. In other words, the generalization of the smaller model $g$ is controlled by hitting probabilities of the initial model $f$ and corrections related to family capacity. The next natural question is the construction of $g$. Inspired by Johnson-Lindenstrauss techniques (cf. also Arora et al. (2018)) we are able to recover the following statement (thorough details are given in Appendix, Subsections B.5, B.6):

**Proposition 5.2** (Informal). *Considering a fully connected feed-forward neural network $f$ where some flatness conditions on the layer decision boundaries are fulfilled, there exists an $\eta$-compression $g$ in the sense of Def. 1 whose number of parameters is logarithmically smaller than $f$.*

Finally, having the generalization estimates on the smaller model $g$ it is natural to attempt transferring those to the initial model $f$ - in Suzuki et al. (2020) this is achieved via certain local Rademacher complexity and "peeling" techniques. However, we choose not to pursue these bounds in the present work and assume the perspective in Arora et al. (2018) that $g$, being almost equivalent to $f$, provides a reasonable indicator of generalization capabilities.

### Acknowledgments

We would like to thank our anonymous reviewers whose advice helped improve the quality of the presentation. We are indebted to Prof. Christian Bauckhage for his constant encouragement, support and fruitful discussions. We also sincerely thank Benjamin Wulff for maintaining the outstanding computation environment at Fraunhofer IAIS - his support and coffee conversations played an essential role for our empirical analysis. In part, this work was supported by the Competence Center for Machine Learning Rhine-Ruhr (ML2R) which is funded by the Federal Ministry of Education and Research of Germany (grant no. 01IS18038B). We gratefully acknowledge this support.

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

## A  APPENDIX A: HITTING ESTIMATES, SATURATION AND CURVATURE

### A.1  BROWNIAN MOTION AND BESSEL PROCESSES

In this Subsection we introduce some basic background on Brownian motion.

**Definition 2** (Brownian motion). *A real-valued stochastic process $\{\omega(t) : t \geq 0\}$ is called a one-dimensional Brownian motion started at $x \in \mathbb{R}$ if the following hold:*

- $\omega(0) = x$,

- *the process has independent increments, that is, for $0 \leq t_1 \leq \cdots t_m$ the increments $\omega(t_j) - \omega(t_{j-1})$ for $j = 2, \cdots, m$ are independent random variables,*

- *for $t \geq 0, h > 0$, the increments $\omega(t + h) - \omega(t)$ are normally distributed with expectation zero and variance $h$,*

- *almost surely, the function $t \mapsto \omega(t)$ is continuous.*

*The process $\{\omega(t) : t \geq 0\}$ is called a standard Brownian motion if $x = 0$.*

*Finally, if $\omega_1, \cdots, \omega_n$ are independent one-dimensional Brownian motions started at $x_1, \cdots, x_n$ then the stochastic process $\omega(t) = (\omega_1(t), \cdots, \omega_n(t))$ is called an $n$-dimensional Brownian motion started at $x = (x_1, \cdots, x_n)$.*

**Remark A.1.** *The distribution of the standard $1$-dimensional Brownian motion $\omega(t)$ is normal with mean $\mathbf{0}$ and variance $t$. It follows that the RMSD (root mean squared displacement) of the standard $n$-dimensional Brownian motion is $\sqrt{nt}$.*

**Sampling**  Brownian motion simulation is prescribed directly by Definition 2. Given a step size $s$, number of steps $k$ we sample a Brownian path as

$$\hat{\omega}(k) := \sum_{i=0}^{k} sX_i, \quad X_i \sim N(0, 1). \tag{9}$$

By Definition 2, $\text{Var}[\omega(t)] = t$, hence the sampling $\hat{\omega}$ corresponds to running a Brownian motion for time

$$t = ks^2. \tag{10}$$

In particular, the mean displacement of $\hat{\omega}$ is $s\sqrt{nk}$. In accordance with the main text, Subsection 3.1 and Fig. 2, whenever we need to sample Brownian motion contained within the ball $B(x, r)$ for its lifespan $[0, t]$, we will fix the number of steps $k$ (usually, we set $k = 400$) and adjust the step size $s$ accordingly, so that $r = s\sqrt{nk}$.

**Estimating hitting probabilities**   A straightforward empirical way to estimate Brownian hitting probability $\mathbb{P}_\omega\left[\exists t_0 \in [0, t] | \omega(t_0) \in S\right]$ of a target set $S$ is to evaluate the steps $\hat{\omega}(i), i = 0, \ldots, k$ and check whether $\hat{\omega}(i_0) \in S$ for some $S$. Of course, the precision of this computation depends on the number of sampled Brownian paths $\hat{\omega}$, as well as the step size $s$ and number of steps $k$. Formal statements on convergence and numerical stability could be obtained, e.g. by means of concentration/Monte-Carlo results (e.g. Proposition B.12 below); however, in practice, in our experiments we mostly worked with the regime $k \approx 10^4$ which seemed an acceptable choice in terms of numeric stability and performance.

Explicit closed-form computation of hitting probabilities is a non-trivial task, though it is possible for some model cases (main text, Lemma 3.2). Dimension 1 is special, where we have the so-called "reflection principle", which says that

$$\mathbb{P}\left(\sup_{0 \le s \le t} \omega(s) \ge d\right) = 2\,\mathbb{P}\left(\omega(t) \ge d\right). \tag{11}$$

For a proof of this basic statement we refer to Mörters & Peres (2010).

However, in higher dimensions, there is no straightforward analog of the reflection principle, and calculating hitting probabilities of spheres leads one to the deep theory of Bessel processes. Let us consider a Brownian particle $\omega(t)$ starting at the origin in $\mathbb{R}^n$ and look at the real-valued random variable $\|\omega(t)\|$ (in the literature, these are known as Bessel processes). We are interested in the probability of the particle hitting a sphere $\{x \in \mathbb{R}^n : \|x\| = r\}$ of radius $r$ within time $t$. Curiously, it seems that there is no known closed formula for such a hitting probability. The only formula we know of is in the form of a convergent series involving zeros of the Bessel function of the first kind, and appears in Kent (1980). For the reader interested in Kent's formula, we also refer to associated asymptotics of zeros of the Bessel function in Watson (1944).

The following heuristic is implicit in many of our calculations and motivates several of our definitions: the probability

$$\mathbb{P}\left(\sup_{0 \le s \le t} \|\omega(s)\| \ge r\right) \tag{12}$$

of a Brownian particle hitting a sphere of radius $r$ within time $t$ is dependent only the ratio $r^2/t$. As a consequence, given a small $\eta > 0$ and a constant $c$, one can choose the constant $c_n$ in $t = c_n r^2$ small enough (depending on $\eta$) such that

$$\mathbb{P}\left(\sup_{0 \le s \le c_n r^2} \|\omega(s)\| \ge cr\right) < \eta. \tag{13}$$

Roughly what this means is the following: for a Brownian particle, the probability of hitting even a large and nearby object may be made arbitrarily small if the motion is not allowed to run sufficiently long.

## A.2   HEAT DIFFUSION AND BROWNIAN MOTION DUALITY

**Macroscopic vs microscopic**   There are roughly two broad viewpoints towards the understanding of diffusion: the "macroscopic" and the "microscopic". Macroscopically, the mechanism of diffusion can be thought of as creating a flux in the direction from greater to lesser concentration. If $u(x, t)$ measures the intensity of the quantity undergoing diffusion, and $J$ the flux across the boundary of a region $\Omega$, then in the simplest model one assumes that (up to a constant) $J = -\nabla u$. Further, we have the identity

$$\partial_t \int_\Omega u(x, t)\, dx = -\int_{\partial\Omega} \nu. - \nabla u\, dS, \tag{14}$$

where $\nu$ is the outward pointing unit normal vector to $\partial\Omega$. By applying the divergence theorem to (14), one immediately gets the heat equation $\partial_t u = \Delta u$. Here $\Delta$ denotes the Laplace operator given by the sum of second derivatives: $\Delta = \sum_{i=1}^{n} \partial_{ii}^2$.

Now, many real-life diffusion processes are the result of microscopic particles jittering around seemingly in a random manner. This motivates the microscopic viewpoint, i.e., the modelling of heat diffusion via Brownian motion of particles. We posit that a particle located at $x \in \mathbb{R}^n$ at time $t_0$ will have the probability $\psi_U(x, t)$ of being in an open set $U \subset \mathbb{R}^n$ at time $t_0 + t$, where

$$\psi_U(x, t) = \int_U p(t, x, y) \, dy, \tag{15}$$

and $p(t, x, y)$ is the fundamental solution of the heat equation, or more famously, the "heat kernel". In other words, $p(t, x, y)$ solves the heat equation

$$\begin{cases} (\partial_t - \Delta) \, u(x, t) = 0, \\ u(x, 0) = \delta(x - y), \end{cases} \tag{16}$$

with the Dirac delta distribution as the initial condition. Via Fourier transform, it is easy to establish that $p(t, x, y)$ is given by

$$p(t, x, y) = \frac{1}{(4\pi t)^{n/2}} e^{-\frac{|x-y|^2}{4t}}. \tag{17}$$

This builds the bridge to pass between analytic statements on the side of the heat equation and probabilistic statements on the side of Brownian motion (see Grigor'Yan (2001), Taylor (2011)). The precise formulation of this duality is given by the celebrated Feynman-Kac theorem discussed in Subsection A.3 below.

**Heating up the decision boundary**    In our context we introduce the following heat diffusion process along the classifier's decision boundary $\mathcal{N}$:

$$\begin{cases} (\partial_t - \Delta) \, \psi(x, t) = 0, \\ \psi(x, 0) = 0, \quad \forall x \in \mathbb{R}^n, \\ \psi(x, t)|_{x \in \mathcal{N}} = 1, \quad \forall t > 0. \end{cases} \tag{18}$$

In other words $\psi(x, t)$ gives the heat quantity at the point $x$ at time $t$ given that at the initial moment $t = 0$ all points have a heat quantity $0$ and afterwards a constant heat source of intensity $1$ is applied only at the decision boundary $\mathcal{N}$. As remarked above this is the macroscopic picture: the mentioned Feynman-Kac duality implies that $\psi(x, t)$ is also the hitting probability $\mathbb{P}_\omega \left[ \exists t_0 \in [0, t] | \omega(t_0) \in \mathcal{N} \right]$.

### A.3    THE FEYNMAN-KAC THEOREM

It is well-known that given a reasonable initial condition $u(x, 0) = f(x)$, one can find an analytic solution to the heat equation via convolution with heat kernel,

$$e^{t\Delta} f(x) := p(t, x, .) * f(.).$$

This just follows from (16) by convolving directly. Now, via the duality of diffusion explained above, one expects a parallel statement on the Brownian motion side, one which computes the contribution of all the heat transferred over all Brownian paths reaching a point at time $t$. It stands to reason that to accomplish this, one needs an integration theory defined over path spaces, which leads us to the theory of Wiener measures. We describe the main idea behind Wiener measure briefly: consider a particle undergoing a random motion in $\mathbb{R}^n$ (given by a continuous path $\omega : [0, \infty) \to \mathbb{R}^n$) in the following manner: given $t_2 > t_1$ and $\omega(t_1) = x_1$, the probability density for the location of $\omega(t_2)$ is

$$p(t, x, x_1) = \frac{1}{(4\pi(t_2 - t_1))^{n/2}} e^{-\frac{|x-x_1|^2}{4(t_2-t_1)}}.$$

We posit that the motion of a random path for $t_1 \le t \le t_2$ is supposed to be independent of its past history. Thus, given $0 < t_1 < \cdots < t_k$, and Borel sets $E_j \subseteq \mathbb{R}^n$, the probability that a path starting at $x = 0$ at $t = 0$, lies in $E_j$ at time $t_j$ is

$$\int_{E_1} \cdots \int_{E_k} p(t_k - t_{k-1}, x_k, x_{k-1}) \cdots p(t_1, x_1, 0) \, dx_k \cdots dx_1.$$

The aim is to construct a countably-additive measure on the space of continuous paths that will capture the above property. The above heuristic was first put on a rigorous footing by Norbert Wiener.

Using the concept of Wiener measure, one gets the probabilistic (microscopic) description of heat diffusion, which is the content of the celebrated Feynman-Kac theorem:

**Proposition A.2.** *Let $\Omega \subseteq \mathbb{R}^n$ be a domain, with or without boundary (it can be the full space $\mathbb{R}^n$). In case of a boundary, we will work with the Laplacian with Dirichlet boundary conditions. Now, let $f \in L^2(\Omega)$. Then for all $x \in \Omega$, $t > 0$, we have that*

$$e^{t\Delta} f(x) = \mathbb{E}_x \left( f\left( \omega(t) \right) \phi_\Omega(\omega, t) \right), \tag{19}$$

*where $\omega(t)$ denotes an element of the probability space of Brownian paths starting at $x$, $\mathbb{E}_x$ is the expectation with regards to the Wiener measure on that probability space, and*

$$\phi_\Omega(\omega, t) = \begin{cases} 1, & if\, \omega([0,t]) \subset \Omega \\ 0, & otherwise. \end{cases}$$

For a more detailed discussion, see Georgiev & Mukherjee (2018a).

### A.4 ISOPERIMETRIC AND ISOCAPACITORY RESULTS

**Isoperimetric bounds** Isoperimetric inequalities relating the volume of a set to the surface area of its boundary have given rise to a wealth of results Burago & Zalgaller (1988). Given a set $M$ with boundary $\partial M$, the basic pattern of isoperimetric inequalities is:

$$\mathrm{Vol}(M) \leq c_1\, \mathrm{Area}(\partial M)^{\frac{n}{n-1}}, \tag{20}$$

where $c_1$ is an appropriate positive constant depending on the dimension $n$. In many cases, equality (or saturation in the sense of almost equality) in (20) is characterized by rather special geometry. For example, classical isoperimetric results answer the question, which planar set with a given circumference possesses the largest area, with the answer being the disk. As discussed in the main text, isoperimetric considerations have recently lead to significant insights about decision boundaries of classifiers subject to adversarial defense training mechanisms Ford et al. (2019) by revealing flattening phenomena and relations to robustness.

**Isocapacitory bounds** As mentioned in the main text, one can prove types of isocapacitory bounds that resemble the isoperimetric ones: roughly speaking, these replace the area term with suitable Brownian hitting probabilities. We have the following result (cf. also Georgiev & Mukherjee (2018a)):

**Proposition A.3.** *Let $B(x,r) \subset \mathbb{R}^n, n \geq 3$, and let $E \subset B(x,r)$ denote an "obstacle", and consider a Brownian particle started from $x$. Then the relative volume of the obstacle is controlled by the hitting probability of the obstacle:*

$$\frac{\mathrm{Vol}(E)}{\mathrm{Vol}(B(x,r))} \leq c_n \left( \psi_E(x,t) \right)^{\frac{n}{n-2}}. \tag{21}$$

*Here, $c_n$ is a positive constant whose value is dependent only on $n$ provided the ratio between $r^2$ and $t$ is suitably bounded. In particular, in the regime $r^2 = nt$, we have that $c_n = \left( \Gamma\left(\frac{n}{2} - 1\right) / \Gamma\left(\frac{n}{2} - 1, \frac{n}{4}\right) \right)^{\frac{n}{n-2}}$. Here, $\Gamma(s, x)$ represents the upper incomplete Gamma function*

$$\Gamma(s, x) := \int_x^\infty e^{-t} t^{s-1}\, dt.$$

*Proof.* Recall that the capacity (or more formally, the 2-capacity) of a set $K \subset \mathbb{R}^n$ defined as

$$\mathrm{Cap}(K) = \inf_{\eta|_K \equiv 1, \eta \in C_c^\infty(\mathbb{R}^n)} \int_{\mathbb{R}^n} |\nabla \eta|^2. \tag{22}$$

From Section 2.2.3, Maz'ya (2011), we have the following "isocapacitory inequality":

$$\mathrm{Cap}(E) \geq \omega_n^{2/n} n^{\frac{n-2}{n}} (n-2)|E|^{\frac{n-2}{n}}, \tag{23}$$

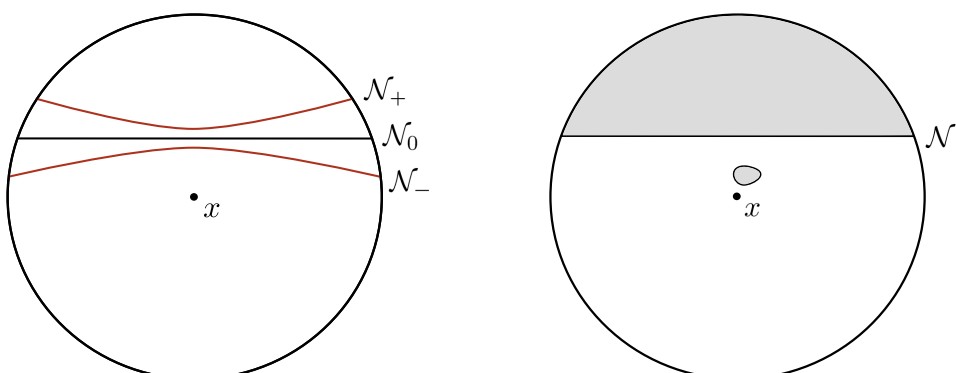

Figure 5: Examples illustrating the interplay between isoperimetric and isocapacitory saturation in high dimensions. *(Left)* Slightly bending a flat decision boundary $\mathcal{N}_0$ causes significant changes in $\tau$ with the isoperimetric inequality still being very close to optimal: $\mathcal{N}_+$ (resp. $\mathcal{N}_-$) leads to a increase (resp. decrease) in $\tau$ (cf. also Fig. 6). *(Right)* Small "pockets" near the data sample $x$ can also cause large Brownian hitting probabilities (hence, large $\tau$ values) with still well-saturated isoperimetric bounds.

where $\omega_n = \frac{2\pi^{n/2}}{\Gamma\left(\frac{n}{2}\right)}$ is the $(n-1)$-dimensional surface area of $S^{n-1}$. Now, we bring in the following estimate given by Theorem 3.7 of Grigor'Yan & Saloff-Coste (2002):

$$\psi_E(x,t) \geq \text{Cap}(E) \int_0^t \inf_{y \in \partial E} p(s,x,y) \, ds. \tag{24}$$

Now, we have

$$\psi_E(x,t) \geq \omega_n^{2/n} n^{\frac{n-2}{n}} (n-2)|E|^{\frac{n-2}{n}} \int_0^t \frac{1}{(4\pi s)^{n/2}} \inf_{y \in \partial E} e^{-\frac{|x-y|^2}{4s}} \, ds$$

$$\geq \omega_n^{2/n} n^{\frac{n-2}{n}} (n-2)|E|^{\frac{n-2}{n}} \int_0^t \frac{1}{(4\pi s)^{n/2}} e^{-\frac{r^2}{4s}} \, ds$$

$$= \omega_n^{2/n} n^{\frac{n-2}{n}} (n-2)|E|^{\frac{n-2}{n}} \frac{1}{4r^{n-2}\pi^{n/2}} \int_{\frac{r^2}{4t}}^{\infty} e^{-z} z^{n/2-2} \, dz.$$

After rearrangement the proposed claim follows. $\qquad\square$

Intuitively, it makes sense that if the volume of a set is fixed, one can increase its hitting probability by "hammering" the set into a large thin sheet. However, it seems unlikely that after lumping the set together (as in a ball), one can reduce capacity/hitting probability any further. Moreover, isocapacitory bounds are saturated by the $n$-ball.

It is also illustrative to compare the seemingly allied concepts of capacity and surface area. A main difference of capacity with surface area is the interaction of capacity with hitting probabilities. As an illustrative example, think of a book which is open at an angle of $180°, 90°, 45°$ respectively. Clearly, all three have the same surface area, but the probability of a Brownian particle striking them goes from the highest to the lowest in the three cases respectively. It is rather difficult to make the heuristic precise in terms of capacity (at least from the definition). Capacity can be thought of as a soft measure of how "spread out" or "opened-up" a surface is, and is highly dependent on how the surface is embedded in the ambient space.

**Isocapacitory vs isoperimetric saturation** A main line of analysis in the present work addresses the interplay between isocapacitory and isoperimetric saturation. In our particular context of defense training mechanisms we observe saturation of isoperimetric bounds for the classifier's decision boundaries - this implies that decision boundaries are not far from being flat. However, as mentioned before, it turns out that isocapacitory saturation does not concentrate around the values corresponding to hyperplanes (overall, it seems to stay well below that value). In this sense, isocapacitory saturation

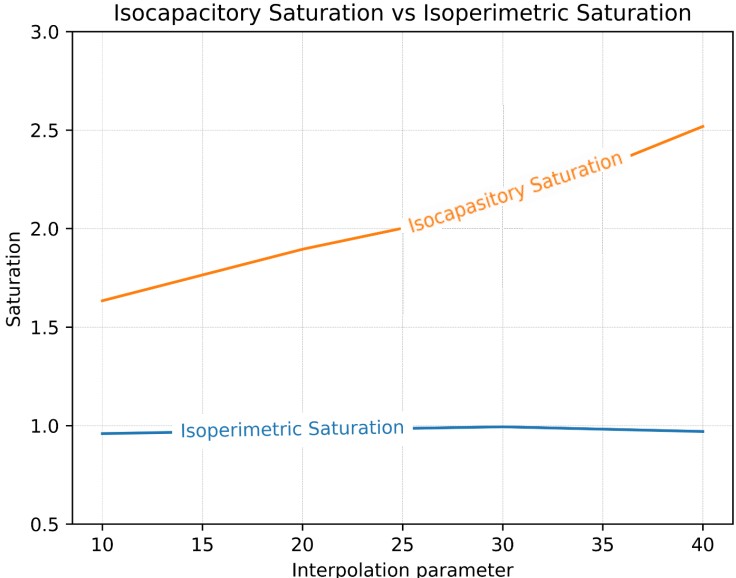

Figure 6: A continuation on Fig. 5: Isocapacitory and isoperimetric saturation while slightly bending the decision boundary ($\mathcal{N}_-$ and $\mathcal{N}_+$ in Fig. 5). In this plot the decision boundary $\mathcal{N}_-, \mathcal{N}_+$ is a cap of a larger sphere with radius $R$ (set initially to $15r$) in dimension $3072$ (corresponding to CIFAR10). We interpolate between $\mathcal{N}_-$ and $\mathcal{N}_+$: first, by increasing the radius $R$, $\mathcal{N}_-$ converges to the flat $\mathcal{N}_0$ and, similarly, starting from $\mathcal{N}_0$ we decrease $R$ to get to $\mathcal{N}_+$. Along this interpolation process, we plot the graphs of the isocapacitory and isoperimetric saturation. In particular, we observe at least $96\%$ saturation of the isoperimetric bound whereas the isocapacitory bounds shows a much more sensitive behaviour on this scale.

acts as a finer sensitive measure of deviation from flatness. A simple model geometric scenario that provides similar behaviour is illustrated in Fig. 5 and Fig. 6.

### A.5 Model Cases

We first begin with the proof of Lemma 3.2.

*Proof.* Let us select an orthonormal basis $\{e_1, \ldots, e_n\}$ so that $e_1$ coincides with the given hyperplane's normal vector. A standard fact about $n$-dimensional Brownian motion is that the projections on the coordinate axes are again one-dimensional Brownian motions Mörters & Peres (2010). Thus, projecting the $n$-dimensional Brownian motion onto $e_1$ the hitting probability of the hyperplane is the same as the probability that one-dimensional Brownian motion $\omega(t)$ will pass a certain threshold $d$ by time $t$. To compute this probability we use the reflection principle (11) in conjunction with Remark A.1. Consequently, the RHS is equal to $2\Phi(-d/\sqrt{t})$. The computation of $\mu(x, r)$ follows by definition. $\square$

Here we note that the dimension $n$ enters only in terms of the spherical cap volume. An impression how $\tau$ behaves for different choices of $n$ in terms of the distance $d$ is given in Fig. 7. In particular, one observes the well-known *concentration of measure* phenomenon and Levy's lemma: the volume of the spherical cap exhibits a very rapid decay as $n$ becomes large. Moreover, experiments reveal a curious phenomenon: there is a threshold distance $d_0$ until which $\tau \approx 2$ and afterwards $\tau$ explodes.

In Fig. 8 we plot further interesting model cases where the error set forms a wedge (the region between two intersecting hyperplanes) or a cone.

**Spiky sets** As discussed in the main text, one observes a high isocapacitory saturation $\tau$ for the so-called "spiky" sets - these are sets of relatively small volume and relatively large/dense boundary.

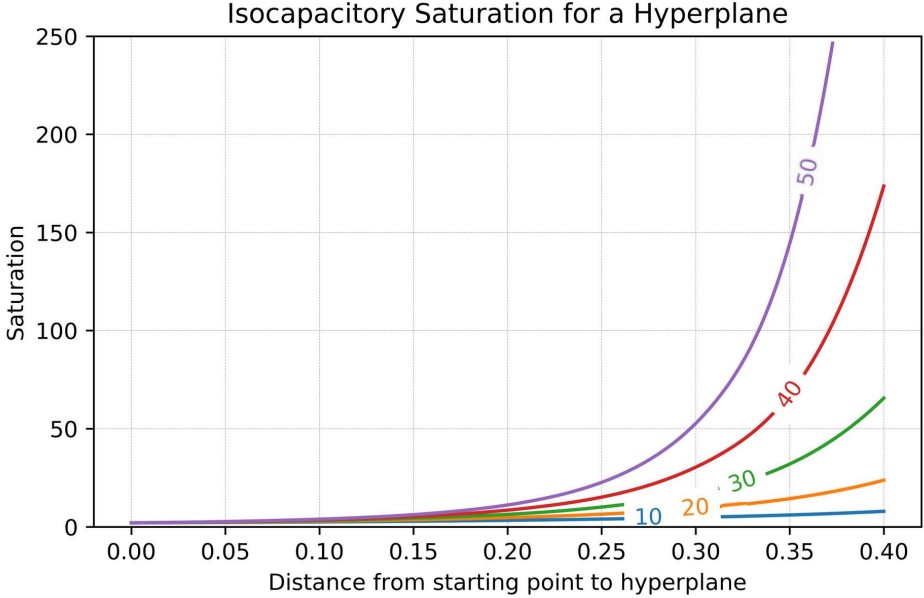

Figure 7: The isocapacitory saturation $\tau$ of a flat error set. Given a point $x$, the computation takes place in $B(x,r)$ with $r = 1$. The distance to the flat decision hyperplane is given on the x-axis, while the y-axis gives $\tau$. Curve labeling indicates the respective dimension. There appears to be a threshold dividing between the regimes $\tau \approx 2$ and $\tau \to \infty$.

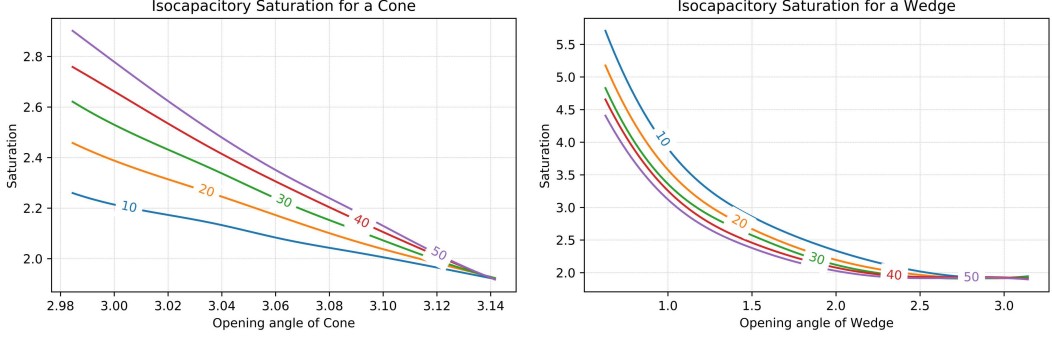

Figure 8: Further model cases and plots of the isocapacitory saturation $\tau$. *(Left)* Isocapacitory saturation of cone in terms of the opening angle (radians). *(Right)* Isocapacitory saturation of wedge in terms of the opening angle (radians). Curve labels indicate the respective dimension. Again one observes concentration of measure as the volume of the cone decreases to 0 exponentially fast in terms of the dimension $n$: this is why we plot the opening angle around $\pi$ in this case. Furthermore, cones and wedges with an opening angle of almost $\pi$ behave like hyperplanes in terms of saturation.

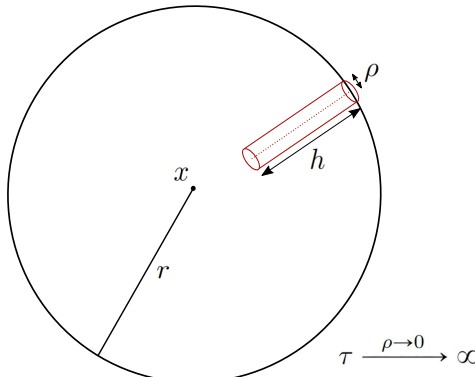

Figure 9: Cylindrical "spike" of height $h$ and radius $\rho$ inside the ball $B(x, r)$.

Theoretically, a guiding model case in this direction is given by Lemma 3.3 in the main text, whose proof we now record.

*Proof.* Let $T_\rho$ denote the $\rho$- tubular neighborhood of a line segment of length $h$ inside $\mathbb{R}^n$. Clearly, $T_\rho \cong B(0, \rho) \times [0, h]$, where $B(0, r)$ is a $\rho$-ball inside $\mathbb{R}^{n-1}$.

By the well-known process of Steiner symmetrization in $\mathbb{R}^n$, it is clear that the expression for capacity in (22) will be minimized by a function that is "radially symmetric" around the central axis of the tube $T_\rho$, that is $f(x, y) = f(|x|)$, where $x \in B(0, \rho), y \in [0, h]$. Then, as we scale $\rho \to \lambda\rho$, where $\lambda \searrow 0$, $\mathrm{Cap}\,(T_{\lambda\rho}) \sim \lambda^{n-3} \mathrm{Cap}\,(T_\rho)$ (which is seen directly from the definition (22)), whereas the volume scales as $|T_{\lambda\rho}| = \lambda^{n-1} |T_\rho|$.

Now assume that the cylinder $T_\rho$ is inside the closed ball $\overline{B(x, r)} \subset \mathbb{R}^n$, the central axis of $T_\rho$ is pointing towards $x$, and $T_\rho$ is touching the boundary of $B(x, r)$. To pass from capacity to hitting probability of the set $T_\rho$, we use that Grigor'Yan & Saloff-Coste (2002):

$$\frac{\mathrm{Cap}(T_\rho)r^2}{\mathrm{Vol}(B(x, r))}e^{-C\frac{r^2}{t}} \leq \psi_{T_\rho}(x, t). \tag{25}$$

Finally, using the definition of $\tau$ and putting the above estimates together, one sees that in the time regime of $O(r^2)$, $\tau$ scales like $\lambda^{-2/(n-2)}$, and hence, $\tau \nearrow \infty$ as $\lambda \searrow 0$.

$\square$

See also Figure 8 for a visual discussion of the isocapacitory saturation for the model cases of wedges and cones.

### A.6  CURVATURE ESTIMATES IN TERMS OF ISOCAPACITORY SATURATION

The geometric concept of curvature has a rich history and plays a central role in differential geometry and geometric analysis. There are several notions of curvature in the literature, ranging from intrinsic notions like sectional, Ricci or scalar curvatures to extrinsic (that is, dependent on the embedding) notions like principal curvatures and mean curvature, which are encoded in the second fundamental form. In this note we use a somewhat "soft" definition of curvature, following previous work Fawzi et al. (2016); Dezfooli et al. (2018). Suppose the decision boundary $\mathcal{N}_f$ is sufficiently regular ($C^2$ is enough for our purpose) and it separates $\mathbb{R}^n$ into two components $\mathcal{R}_1 := \{f > 0\}$ and $\mathcal{R}_2 := \{f < 0\}$, corresponding to a binary classification (the construction in the multi-label case is analogous). For a given $p \in \mathcal{N}_f$, let $r_j(p)$ denote the radius of the largest sphere that is tangent to $\mathcal{N}_f$ at $p$, and fully contained in $\mathcal{R}_j$. Then, one defines the curvature $\kappa$ at $p$ as

$$\kappa(p) = 1/\min\left(r_1(p), r_2(p)\right). \tag{26}$$

See Fig. 10 for a geometric illustration. However, it turns out that most notions of curvature are quite subtle (see Fawzi et al. (2016)) and at this point, seemingly more cumbersome and intractable to

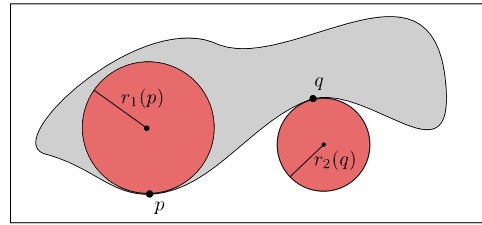

Figure 10: "Soft" definition of curvature given by the inverse radius of the osculating sphere.

handle experimentally. We will take an indirect approach, and attempt to read off the effect of and on curvature via the isocapacitory saturation $\tau$.

Again, we begin with the model cases: we first study the behaviour of curvature $\kappa$ if $\tau$ achieves its least possible value. We start by fixing some notation. As before let us consider a ball $B(x, r)$ with an error set $E \subset B(x, r)$ and boundary $\mathcal{N} = \partial E$ (clearly our main case of interest is $E = E(y) \cap B(x, r)$). Let us denote the the distance $d = d(x, \mathcal{N})$ and suppose the point $y \in \mathcal{N}$ realizes this distance, i.e. $d(x, y) = d$. To rule out some degenerate cases and ease the analysis we introduce the following assumption:

**Assumption**: The hypersurface $\mathcal{N}$ and the point $x$ are on different sides of the tangent hyperplane $H^* := T_y \mathcal{N}$ (cf. Fig. 11).

This assumption is also technically important, as otherwise low values of $\tau$ will be produced by annuli surrounding $x$. With that in place, we have the following rigidity result:

**Proposition A.4.** *Let us fix the distance $d = d(x, \mathcal{N})$ and suppose the assumption above holds. Then the least possible value of $\tau$ is attained only if the curvature $\kappa$ of the hypersurface $\mathcal{N}$ is 0.*

*Proof.* As above let $H^*$ be the tangent hyperplane at distance $d$ from $x$, and let $C$ denote the (smaller) spherical cap formed by $H^* \cap B(x, r)$. The proof relies on the following variational argument. If $\mathcal{N}$ is not the same as $H^*$, then $\mathcal{N} \subseteq C$, with $y \in \mathcal{N} \cap H^*$. We wish to argue then one can perturb $\mathcal{N}$ infinitesimally to decrease the value of $\tau$, so the only minimizer of the above expression has to be $H^*$. The basic idea is to cut out a small piece $p_v$ around $v$ and paste it in the region of around $\tilde{v}$ (Fig. 11).

We say that $\mathcal{N}$ has positive curvature at some point $z$ if the ball defining the curvature at $z$ and the point $x$ lie on different sides of $\mathcal{N}$. The construction is as follows. Let $S(x, s)$ be the $(n - 1)$-sphere centered at $x$ with radius $s$. We consider two cases:

**Case I**: Let us suppose that there exist $s_1 < s_2 \leq r$ and points $v, \tilde{v} \in \mathcal{N}$ such that the curvature of $\mathcal{N}$ at $v \in \mathcal{N} \cap S(x, s_1)$ is greater than the curvature at $\tilde{v} \in \mathcal{N} \cap S(x, s_2)$. Let us, moreover, choose the infimum among such $s_1$ and the supremum among such $s_2$.

To define the mentioned piece $p_v$, we consider two small balls $B(v, \varepsilon), B(\tilde{v}, \varepsilon)$ (where $\varepsilon \ll s_2 - s_1$), and cut out a set $p_v = E \cap B(v, \varepsilon)$ such that $\partial (E \setminus B(v, \varepsilon))$ is congruent to $\mathcal{N} \cap B(\tilde{v}, \varepsilon)$ (this is possible due to the curvature assumptions at $v, \tilde{v}$). Then, we define the new error set $E' = E \cup p_{\tilde{v}} \setminus p_v$ and the boundary $\mathcal{N}' = \partial E'$, where $p_{\tilde{v}}$ represents the image of $p_v$ under the rigid motion and attached inside $B(\tilde{v}, \varepsilon)$ (see Fig. 11). It is now clear that $|E| = |E'|$, but $\psi_{E'}(x, T) < \psi_E(x, T)$ for all $T > 0$. The last inequality follows from the evaluation of the explicit heat kernel that defines hitting probability $\psi$ as stated by Feynman-Kac duality:

$$
\begin{aligned}
\psi_E(x, T) &= \int_0^T \int_E \frac{1}{(4\pi t)^{n/2}} e^{-\frac{(x-y)^2}{4t}} \, dy \, dt \\
&> \int_0^T \int_{E'} \frac{1}{(4\pi t)^{n/2}} e^{-\frac{(x-y)^2}{4t}} \, dy \, dt = \psi_{E'}(x, T).
\end{aligned}
$$

It follows from the definition of $\tau$ that $\tau_E \geq \tau_{E'}$.

**Case II**: If Case I is not satisfied, then, similarly, we choose two points $v, \tilde{v}$, but instead of defining the piece $p_v$ by intersection with a small ball around $v$ we select $p_v$ as a "concavo-convex lens shape" domain, where the curvature on the concave "inner side" of $p_v$ of the lens is greater than that on the

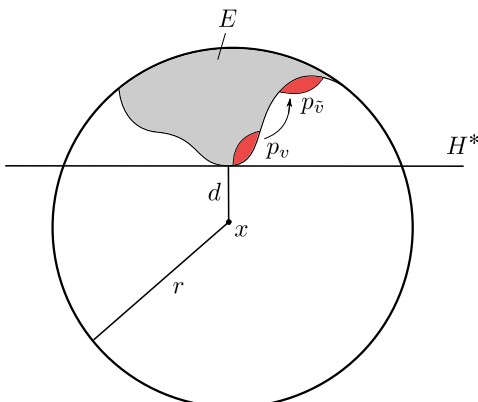

Figure 11: Moving the piece $p_v$ near the tip of the obstacle and reattaching it far away as $p_{\tilde{v}}$ reduces the hitting probability, but preserves volume.

convex outer side. As before, we attach a rigid motion image of $p_v$ inside $B(\tilde{v}, \varepsilon)$. The rest of the argument is similar to Case I. $\qquad\square$

With reference to our previous discussion of spikes, it heuristically makes sense that a spike must have reasonably high curvature (it can have high curvature on the average, or if it is flat at most places, then have a sharp needle like end where the curvature is very high). In the same setting as Proposition A.4 let us, moreover, for simplicity assume that $\mathcal{N}$ is the graph of a function over the tangent hyperplane $H^*$ (Fig. 11).

**Proposition A.5.** *In the above setting let us fix the value of* $d$. *Then, if the maximum curvature* $\kappa_{\max}$ *of* $\mathcal{N}$ *is sufficiently high (greater than some universal constant), then it satisfies*

$$\kappa_{\max} \geq \frac{\tau^{\frac{1}{n}}}{r} \left( \Phi\left( -\frac{d}{\sqrt{t}} \right) \right)^{-\frac{1}{n-2}}, \tag{27}$$

*where* $\Phi$ *denotes the c.d.f. of the standard normal distribution. If a point attaining this maximum curvature is within the half concentric ball* $B(x, r/2)$, *then* $\kappa_{\max}$ *satisfies the stronger estimate*

$$\kappa_{\max} \geq \frac{\tau^{\frac{1}{n}}(r-d)}{r^{\frac{n}{n-1}}} \left( \Phi\left( -\frac{d}{\sqrt{t}} \right) \right)^{-\frac{n}{(n-1)(n-2)}}. \tag{28}$$

*Proof.* Recalling the definition of the isocapacitory saturation $\tau$, we will bound the numerator (resp. denominator) of $\tau$ from above (resp. below). First, for the numerator $\psi_E(x, t)$ we will use a basic monotonicity property of hitting probabilities stating that for two sets $A \subseteq B$ one has $\psi_A(x, t) \leq \psi_B(x, t)$ - this follows directly from the definition of $\psi$. Now, since $E \subseteq C$ where $C$ is the smaller spherical cap of $B(x, r) \cap H^*$, we have $\psi_E(x, t) \leq \psi_C(x, t)$. However, recalling the explicit form of $\psi_C$ from Lemma 3.2 of the main text, we have

$$\psi_E(x, t) \leq \Phi\left( -\frac{d}{\sqrt{t}} \right).$$

Second, to bound the denominator of $\tau$ (i.e. $\text{Vol}(E)$), we observe that if $\kappa_{\max}$ is large enough, by definition $E$ contains a ball of radius $\frac{1}{\kappa_{\max}}$, and $\text{Vol}(E) \geq \frac{\omega_n}{\kappa_{\max}^n}$ where $\omega_n$ denotes the volume of unit $n$-dimensional ball. That finally implies,

$$\tau \leq \left( \Phi\left( -\frac{d}{\sqrt{t}} \right) \right)^{\frac{n}{n-2}} \frac{\text{Vol}(B(x, r))}{\text{Vol}(E)}$$

$$\leq \left( \Phi\left( -\frac{d}{\sqrt{t}} \right) \right)^{\frac{n}{n-2}} r^n \kappa_{\max}^n,$$

which proves (27).

If a point of maximum curvature is inside a concentric ball of radius $r/2$, then $E$ contains $\approx \frac{\kappa_{\max}(r-d)}{2}$ balls of radius $\frac{1}{\kappa_{\max}}$, which implies that $\mathrm{Vol}(E) \geq \kappa_{\max}(r-d)\left(\frac{\omega_n}{\kappa_{\max}^n}\right)$.

The rest of the proof is similar. $\qquad\square$

Now, we give a curvature estimate which works in any regime, without any restrictions. The tradeoff is a global average bound of the $L^p$-type rather than pointwise estimates.

**Proposition A.6.** *In the setting as above, let us fix the distance $d = d(x, \mathcal{N})$. At each point of $\mathcal{N}$, let us denote by $\kappa$ the maximal sectional curvature of $\mathcal{N}$ at that point. The following estimate holds:*

$$\|\mathcal{K}\|_{L^1} \geq V_n(d, r) - \frac{2\omega_n r^n \Phi\left(-\frac{d}{\sqrt{t}}\right)}{\tau_H}, \tag{29}$$

*where $V_n(d, r)$ denotes the volume of the smaller spherical cap at distance $d$, the constant $\omega_n$ denotes the volume of unit ball in $\mathbb{R}^n$, and the function $\mathcal{K}$ is an integral function of the curvature $\kappa$ over lines (defined in (31) below).*

*Proof.* Again, we suitably bound the numerator and denominator of $\tau$. Starting with the numerator, as explained in Proposition A.5, we have by monotonicity

$$\psi_E(x, t) \leq 2\Phi\left(-\frac{d}{\sqrt{t}}\right). \tag{30}$$

To bound the denominator of $\tau$ we proceed as follows. Let $\mathcal{N}$ be the graph of the function $\tilde{g}(x_1, \cdots, x_{n-1})$, where the variables $x_j$ are taken from the hyperplane $H^*$ (Fig. 11) at distance $d$ from $x$; the point at which $\mathcal{N}$ touches this hyperplane is taken as the origin. Let $\varphi_\epsilon$ be a smooth cut-off function defined on the hyperplane such that $\varphi \equiv 1$ on the set $S$ of all $(x_1, \cdots, x_{n-1})$ such that $\tilde{g}(x_1, \cdots, x_{n-1}) \in B(x, r)$, and $\varphi \equiv 0$ outside the $\epsilon$-tubular neighborhood of $S$. Finally, let $g_\epsilon := \varphi_\epsilon \tilde{g}$.

Now we see that, letting $a = (r^2 - d^2)^{1/2}$,

$$V_n(d, r) - \mathrm{Vol}(E) \leq \int_{\rho=0}^a \int_{S^{n-2}} g_\epsilon(\rho, \theta)\, \rho^{n-2}\, d\rho\, d\theta.$$

Now, if $\eta$ denotes the unit vector in the direction of a fixed $(\rho, \theta)$, observing that $g_\epsilon(0) = 0$, we have by the fundamental theorem of calculus

$$g_\epsilon(\rho, \theta) = \int_0^1 \partial_t g_\epsilon(t\rho\eta, \theta)\, dt.$$

In turn, applying the fundamental theorem a second time and observing that $\nabla g_\epsilon(0) = 0$, we have that

$$g_\epsilon(\rho, \theta) = \int_0^1 \int_0^1 \partial_s \partial_t g_\epsilon(st\rho\eta, \theta)\, ds\, dt.$$

Putting everything together we get,

$$V_n(d, r) - \mathrm{Vol}(E) \leq \int_{\rho=0}^a \int_{S^{n-2}} \left(\int_0^1 \int_0^1 \partial_s \partial_t g_\epsilon(st\rho\eta, \theta)\, ds\, dt\right) \rho^{n-2}\, d\rho\, d\theta.$$

Now, we define the following integral quantity:

$$\mathcal{K}_\epsilon(\rho, \theta) = \int_0^1 \int_0^1 |\kappa_\epsilon(st\rho\eta, \theta)|\, ds\, dt. \tag{31}$$

Noting that the maximum sectional curvature bounds the second derivatives, finally we have that

$$V_n(d, r) - \mathrm{Vol}(E) \leq \|\mathcal{K}_\epsilon\|_{L^1}. \tag{32}$$

To obtain (29) we now put all the above estimates together and let $\epsilon \searrow 0$. $\qquad\square$

# B   APPENDIX B: GENERALIZATION BOUNDS AND COMPRESSION SCHEMES

**Background**   A main line of ML and statistical inference research addresses questions of generalization. To set the stage we start with some notation. Let us suppose that the dataset $\mathcal{X}$ is sampled from a probability distribution $D$, i.e. $(x, y) \sim D$. Following conventions from the literature Arora et al. (2018) we define the expected margin loss of a classifier $f$ by

$$L_\gamma(f) := \mathbb{P}_{(x,y)\sim D}\left[f(x)[y] \leq \gamma + \max_{j=1,\ldots,k; j\neq y} f(x)[j]\right]. \tag{33}$$

We use the notation $\hat{L}_\gamma$ to denote the expected empirical margin loss over the given data set $\mathcal{X}$. Finally, the generalization error is defined as $L_\gamma - \hat{L}_\gamma$.

Quite roughly speaking, standard generalization results attempt to estimate the performance of the classifier on unseen samples (i.e. the full data distribution), thus yielding bounds of the form:

$$L_{\gamma_1}(f) \leq \hat{L}_{\gamma_2}(f) + F(\gamma_1, \gamma_2, f, \mathcal{X}), \tag{34}$$

where $F$ is an additional term that usually depends, e.g. on the size of $\mathcal{X}$, the expressiveness of $f$ and further margin information $(\gamma_1, \gamma_2)$.

## B.1   COMPRESSION IN A HEAT DIFFUSION SENSE IMPLIES GENERALIZATION BOUNDS

We first state a well-known concentration inequality due to Hoeffding which will find repeated use in the ensuing sections:

**Proposition B.1** (Hoeffding's inequality). *Let $X_1, \ldots, X_n$ be independent random variables taking values in the interval $[0, 1]$, and let $\overline{X} = \frac{1}{n}(X_1 + \cdots + X_n)$ be the empirical mean of these random variables. Then we have:*

$$\mathbb{P}\left(\overline{X} - \mathbb{E}\left(\overline{X}\right) \geq t\right) \leq e^{-2nt^2}. \tag{35}$$

We now provide the proof of Proposition 5.1 of the main text.

*Proof.* The strategy of proof follows well-known "weak-law-of-large-numbers" concentration techniques in a spirit similar to Arora et al. (2018).

**Step 1**. First, we show that for a given $g$ as $|\mathcal{X}| \to \infty$,

$$\mathbb{P}_{(x,y)\sim \mathcal{X}}\left(C_g(x, y, t_1)\right) \to \mathbb{P}_{(x,y)\sim D}\left(C_g(x, y, t_1)\right), \tag{36}$$

where $C_g(x, y, \gamma^2)$ is the event that a Brownian path starting at $x$ hits $E_g(y)$ within time $\gamma^2$. The rate of convergence is determined through Chernoff concentration bounds.

Choose $\alpha \in A$, and let $g_\alpha$ be the corresponding classifier. Attached to each sample point $x_j$, there is a Bernoulli random variable $X_j$ which takes the value 1 if $C_{g_\alpha}(x_j, y, \gamma^2)$ happens, and 0 otherwise. Then, the average $\overline{X} = \frac{1}{m}\sum_{j=1}^m X_j$ is given by the average of $m$ i.i.d. Bernoulli random variables each of whose expectations is given by $\mathbb{P}_{(x,y)\sim D} C_{g_\alpha}(x, y, \gamma^2)$. Furthermore, we note that if a data sample is misclassified, then the Brownian particle almost surely will hit the error set. Combining this observation with the concentration estimate (35) above, we obtain

$$L_0(g_\alpha) \leq \mathbb{P}_{(x,y)\sim D}\left(C_{g_\alpha}(x, y, \gamma^2)\right)$$
$$\leq \mathbb{P}_{(x,y)\sim \mathcal{X}}\left(C_{g_\alpha}(x, y, \gamma^2)\right) + \xi, \tag{37}$$

with probability at least $1 - e^{-2\xi^2 m}$. If each classifier $g_\alpha$ has $q$ parameters, each of which can take $r$ discrete values, we take $\xi = \sqrt{\frac{q\log r}{m}}$.

**Step 2**. The estimate from the previous step should hold for every classifier $g_\alpha$ in the family $A$ with large probability. This is guaranteed by a union bound and tuning the Chernoff bounds from the convergence rate. More precisely, there are $r^q$ different choices $\alpha \in A$, and hence by taking the union of the estimate in (37), one can say that

$$\mathbb{P}_{(x,y)\sim D}\left(C_{g_\alpha}(x, y, \gamma^2)\right) \leq \mathbb{P}_{(x,y)\sim \mathcal{X}}\left(C_{g_\alpha}(x, y, \gamma^2)\right) + \sqrt{\frac{q\log r}{m}} \tag{38}$$

with probability at least $1 - e^{-q \log r}$ over all $\alpha \in A$.

**Step 3**. Finally one uses the fact that $f$ is approximable by at least one $g = g_{\alpha_0}$ for some $\alpha_0$ in $A$. Via Definition 1 of the main text, one sees that

$$\mathbb{P}_{(x,y) \sim \mathcal{X}} \left( C_{g_{\alpha_0}}(x, y, \gamma^2) \right) \leq \mathbb{P}_{(x,y) \sim \mathcal{X}} \left( C_f(x, y, \gamma^2) \right) + \eta,$$

which finally gives that with probability at least $1 - e^{-q \log r}$, we have

$$L_0(g) \leq \mathbb{P}_{(x,y) \sim \mathcal{X}} \left( C_f(x, y, \gamma^2) \right) + \eta + O\left( \sqrt{\frac{q \log r}{m}} \right). \tag{39}$$

$\square$

**Remark B.2.** *As noted, a classifier $f$ classifies a point $x$ wrongly if and only if $\psi_{E(y)}(x, t) = 1$ for all time scales $t$. With this observation, and since (39) works for all real numbers $\gamma$, letting $\gamma \to 0$, we have that with probability at least $1 - e^{-q \log r}$,*

$$L_0(g) \leq \hat{L}_0(f) + \eta + O\left( \sqrt{\frac{q \log r}{m}} \right).$$

*This recovers a loss estimate which is similar to the estimate in Theorem 2.1 of [1].*

*Indeed, one can consider $\mathbb{P}_{(x,y) \sim \mathcal{X}} \left( C_f(x, y, \gamma^2) \right)$ as a "soft" or probabilistic measure of classification with margin $\approx \gamma$.*

When defining the notion of a compression, instead of taking a pointwise difference as in Definition 1 of Arora et al. (2018), we would like to capture the idea that the decision boundary of a good compression should be "close enough" to the decision boundary of the original classifier. In our context, this implies that their "heat signatures" at the sample points should be close enough at all time scales. As noted in the main text, Definition 1 is definitely one natural option to define goodness of compression in a heat-diffusion sense. Another natural way is to consider the Brownian motion's running time and define a good approximation as follows:

**Definition 3.** *Given a positive real number $\eta$, a classifier $g$ is said to be an $\eta-$compression w.r.t. hitting time of $f$ if*

$$\psi_{E_g(y)}(x, \gamma^2 - \eta) \leq \psi_{E_f(y)}(x, \gamma^2) \leq \psi_{E_g(y)}(x, \gamma^2 + \eta) \tag{40}$$

*for all points $x$ in the training sample, labels $y$ and real numbers $\gamma^2 \geq \eta$.*

Analogously, we have the following

**Proposition B.3.** *Let us suppose that $f$ is approximable by $g$ in the sense of Definition 3. Here $g \in A$, where $A$ is a family of classifiers $\mathbb{R}^n \to \mathbb{R}$ parametrized by $q$ parameters assuming $r$ discrete values. As before, for a classifier $h$, let $C_h(x, y, t)$ be the event that a Brownian path starting at $x$ hits $E_h(y)$ within time $t$. Then we have*

$$L_0(g) \leq \mathbb{P}_{(x,y) \sim D} \left( C_{g_\alpha}(x, y, \gamma^2 - \eta) \right) \leq \mathbb{P}_{(x,y) \sim \mathcal{X}} \left( C_f(x, y, \gamma^2) \right) + O\left( \sqrt{\frac{q \log r}{m}} \right) \tag{41}$$

*with probability at least $1 - e^{-q \log r}$.*

The proof proceeds similarly as above. Letting $\gamma^2 \to \eta$ gives us

$$L_0(g) \leq \mathbb{P}_{(x,y) \sim \mathcal{X}} \left( C_f(x, y, \eta) \right) + O\left( \sqrt{\frac{q \log r}{m}} \right). \tag{42}$$

Again, the first term on the RHS can be interpreted as the geometric margin of classification. In particular, if the classifier $f$ separates points by a distance of $\approx \sqrt{n\eta}$, then since the Brownian motion travels $\approx \sqrt{n\eta}$ hitting the error set will happen only if a misclassification occurred, i.e. we have

$$\mathbb{P}_{(x,y) \sim \mathcal{X}} \left( C_f(x, y, \eta) \right) \approx L_0(f). \tag{43}$$

## B.2 A SHARP VARIANT OF THE JOHNSON-LINDENSTRAUSS ALGORITHM

Several state-of-art compression schemes utilize a dimensionality reduction in the spirit of Johnson-Lindenstrauss (JL), Arora et al. (2018). In this Subsection we discuss a JL compression scheme that will later be coupled with and tuned by some heat-diffusion estimates. We begin by discussing a variant of JL (Alg. 1).

---

**Data:** Original matrix $A$ of dimension $h_1 \times h_2$, $\beta \in (0, 1)$.
**Result:** Stochastic compressed matrix $\hat{A}$ with $O\left(\log(h_1 h_2)/\beta\alpha^2\right)$ non-zero entries such that

$$\mathbb{P}\left[\|\hat{A}x - Ax\| \geq \alpha\|A\|_F\|x\|\right] \leq \beta.$$

Start with matrix $A$, real number $\alpha$;
**while** $i \leq h_1$, $j \leq h_2$ **do**

  Let $z_{ij} = 1$ with probability $p_{ij} = \frac{2a_{ij}^2}{\beta\alpha^2\|A\|_F^2}$, 0 otherwise;
  Let $\hat{a}_{ij} = \frac{z_{ij}a_{ij}}{p_{ij}}$.
**end**
Return $\hat{A} = (\hat{a}_{ij})$.

**Algorithm 1:** Compressing a matrix $A \in \mathbb{R}^{h_1 \times h_2}$

---

**Proposition B.4.** *Let $A$ be a matrix of dimension $h_1 \times h_2$. Then, one can find a compressed matrix $\hat{A}$ such that*

$$\|Ax - \hat{A}x\| \leq \alpha\|A\|_F\|x\|,$$

*with probability at least $1 - \beta$, where the number of parameters of $\hat{A}$ is $O\left(\log(h_1 h_2)/\beta\alpha^2\right)$.*

A proof of Proposition B.4 in the spirit of classical JL can be provided - however, here we introduce a Bernoulli scheme which is a minor modification of Algorithm 2 of Arora et al. (2018).

*Proof.* Define the random variables $z_{ij}$ which take the value 1 with probability $p_{ij} = \frac{2a_{ij}^2}{\beta\alpha^2\|A\|_F^2}$, and the value 0 otherwise. Define $\hat{a}_{ij} = \frac{z_{ij}a_{ij}}{p_{ij}}$. One can now calculate that $\mathbb{E}(\hat{a}_{ij}) = a_{ij}$, and $\mathrm{Var}(\hat{a}_{ij}) \leq \beta\alpha^2\|A\|_F^2$. Using the above, one can further calculate that $\mathbb{E}(\hat{A}x) = Ax$, and $\mathrm{Var}(\hat{A}x) \leq \|x\|^2\|A\|_F^2\beta\alpha^2$. By Chebyshev's inequality, this gives us that

$$\mathbb{P}\left[\|\hat{A}x - Ax\| \geq \alpha\|A\|_F\|x\|\right] \leq \beta.$$

Now, the expected number of non-zero entries in $\hat{A}$ is $\sum_{i,j} p_{ij} = \frac{2}{\beta\alpha^2}$. An application of Chernoff bounds now gives that with high probability the number of non-zero entries is $O\left(\log(h_1 h_2)/\beta\alpha^2\right)$. $\square$

## B.3 HITTING PROBABILITY, CAPACITY SENSITIVITY AND COMPRESSION

As discussed in the main text, here we use hitting probabilities associated to the decision boundary to define a concept "capacity sensitivity" of a neural net layer. The heuristic is, the less the capacity sensitivity of a layer, the greater the facility in compressing the layer to one with fewer parameters. This goes in the spirit of current state-of-art results on compression and generalization bounds (Arora et al. (2018), Suzuki et al. (2018), Suzuki et al. (2020)). In particular, in Arora et al. (2018) the authors provide the notions of noise sensitivity and noise cushions motivated by Gaussian noise injections. Our first proposed definition for "heat-diffusion noise cushions" and capacity sensitivity goes as follows:

**Definition 4.** *Let $\eta \sim \mathcal{N}$ be distributed along a noise distribution $\mathcal{N}$ concentrated in ball $\|\eta\| \leq \eta_0$. We define the capacity sensitivity $S(x, A_i; t)$ of a layer $A_i$ at the point $x$ as*

$$S(x, A_i; t) := \mathbb{E}_{\eta \sim \mathcal{N}} \frac{\left|\psi_{E_f}(\phi(A_i(x + \|x\|\eta)), t) - \psi_{E_f}(\phi(A_i x), t)\right|}{\left|\psi_{E_f}(\phi(A_i x), t)\right|}. \tag{44}$$

*We denote the maximum and expected sensitivity respectively as*

$$S^m(A_i; t) := \max_{x \in \mathcal{X}} S(x, A_i; t), \quad S^e(A_i; t) := \mathbb{E}_{x \sim \mathcal{X}} S(x, A_i; t). \tag{45}$$

Now we use Algorithm 1 to investigate a method for compressing a layer $A_i$ so that the capacity properties are preserved.

**Proposition B.5.** *Let a particular layer $A_i$ of the neural net be of dimension $h_1 \times h_2$. Then, Algorithm 1 generates an approximation $\hat{A}_i$ with $O\left(\log(h_1 h_2)/\beta \alpha^2\right)$ parameters for which we guarantee that $\psi_{E_f(y)}(\phi(\hat{A}_i))$ is proportional to $\psi_{E_f(y)}(\phi(A_i))$ up to an error $\epsilon$ with probability $\beta + S^m(A_i)/\epsilon$.*

*Proof.* Using the fact that $\psi_{E_f(y)}\left(\phi\left(\hat{A}x\right), t\right) = \psi_{E_f(y)}\left(\phi\left(A(x + \|x\|\eta)\right), t\right)$, let $A_\delta$ denote the event that

$$\left| \frac{\psi_{E_f(y)}\left(\phi(\hat{A}_i x), t\right) - \psi_{E_f(y)}\left(\phi(A_i x), t\right)}{\psi_{E_f(y)}(\phi(A_i x), t)} \right| = \left| \frac{\psi_{E_f(y)}(\phi(A_i(x + \|x\|\eta)), t) - \psi_{E_f(y)}(\phi(A_i x), t)}{\psi_{E_f(y)}(\phi(A_i x), t)} \right| \geq \delta.$$

For every fixed $x \in \mathcal{X}$, using (44) and Markov's inequality immediately implies

$$\mathbb{P}[A_\delta] \leq \frac{S(x, A_i; t)}{\delta}. \tag{46}$$

Since Algorithm 1 yields controlled distortion, we have that given error parameters $\alpha, \beta$, one gets $\hat{A}$, a stochastic approximation of $A$ such that

$$\mathbb{P}\left[\|\hat{A}_i(x) - A_i(x)\| \geq \alpha \|A_i\|_F \|x\|\right] \leq \beta. \tag{47}$$

Here the reduced number of the parameters of $\hat{A}$ is $O\left(\log(h_1 h_2)/\beta \alpha^2\right)$. With that, we have

$$\mathbb{P}\left[\hat{A}_\delta\right] = \mathbb{P}\left[\left(\frac{\|\hat{A}_i(x) - A_i(x)\|}{\alpha \|A_i\|_F \|x\|} < 1\right) \bigcap \hat{A}_\delta\right] + \mathbb{P}\left[\left(\frac{\|\hat{A}_i(x) - A_i(x)\|}{\alpha \|A_i\|_F \|x\|} \geq 1\right) \bigcap \hat{A}_\delta\right] \tag{48}$$

$$\leq \mathbb{P}[A_\delta] + \mathbb{P}\left[\frac{\|\hat{A}_i(x) - A_i(x)\|}{\alpha \|A_i\|_F \|x\|} \geq 1\right]$$

$$\leq \frac{S(x, A_i; t)}{\delta} + \beta.$$

This concludes the claim. $\qquad\qquad\square$

The above proposition may seem suboptimal and even somewhat of a tautology, but we include all the details, because one way forward is now evidently clear. In particular, the step in (48) can be improved if we know that if the distance between two vectors $z$ and $w$ is bounded above, then $\psi_{E_f}(z, t) - \psi_{E_f}(w, t)$ is bounded above. In plain language, we would like to say the following: if two points are close, then the respective probabilities of Brownian particles starting from them and hitting $\mathcal{N}_f$ are also close. This is too much to expect in general, but can be accomplished when one places, in addition, certain nice assumptions on the decision boundary.

### B.4  PROOF OF FIRST PART OF PROPOSITION 5.2 OF THE MAIN TEXT

We will break down the proof over three propositions, to illustrate the flow of ideas. The first is the case of the hyperplane which we discussed to some extent above in our curvature analysis (see also Lemma 3.2 of the main text).

**Proposition B.6.** *If the decision boundary $\mathcal{N}_f$ is a hyperplane, then given $\beta, \epsilon$, one can find an $\alpha$ for which the compression scheme of Algorithm 1 gives a compression of a layer $A_i$ of dimension $h_1 \times h_2$ to $\hat{A}_i$ with $O\left(\log(h_1 h_2)/\beta \alpha^2\right)$ parameters such that*

$$\mathbb{P}\left[\|A_i(x) - \hat{A}_i x\| \leq \alpha \|A_i\|_F \|x\|\right] \geq 1 - \beta,$$

*and*

$$\left| \psi_{E_f}(A_i x, t) - \psi_{E_f}(\hat{A}_i x, t) \right| \le \epsilon$$

*with probability at least* $1 - \beta$. *Here* $t = O\left(dist(A_i(x), \mathcal{N}_f)^2\right)$. *The choice of* $\alpha$ *is made explicit by (50) below.*

*Proof.* Let $w, z \in \mathbb{R}^n$ be two points such that $\|w - z\| \le \delta$. It is clear that the maximum value of $\left| \psi_{E_f}(w, t) - \psi_{E_f}(z, t) \right|$ is given by the probability that a Brownian particle starting from a point $x \in \mathbb{R}^n$ strikes a "slab" of thickness $\delta$ at a distance $d - \delta$ from $x$ (a slab is a tubular neighborhood of a hyperplane) within time $t$. Without loss of generality, assume that the point $z$ is at a distance $d$ from the hyperplane $\mathcal{N}_f$. Then,

$$0 \le \left| \psi_{E_f}(w, t) - \psi_{E_f}(z, t) \right| \le 2 \left( \Phi\left( -\frac{d - \delta}{\sqrt{t}} \right) - \Phi\left( -\frac{d}{\sqrt{t}} \right) \right),$$

which implies that

$$\left| \frac{\psi_{E_f}(w, t) - \psi_{E_f}(z, t)}{\psi_{E_f}(z, t)} \right| \le 2 \left( \frac{\Phi\left( -\frac{d - \delta}{\sqrt{t}} \right)}{\Phi\left( -\frac{d}{\sqrt{t}} \right)} - 1 \right).$$

From the above calculation, we get that

$$\frac{\|A_i(x) - \hat{A}_i(x)\|}{\|A_i\|_F \|x\|} \le \alpha \implies \frac{\left| \psi_{E_f}(A_i(x), t) - \psi_{E_f}(\hat{A}_i(x), t) \right|}{\left| \psi_{E_f}(A_i(x), t) \right|} \le 2 \left( \frac{\Phi\left( -\frac{d - \delta}{\sqrt{t}} \right)}{\Phi\left( -\frac{d}{\sqrt{t}} \right)} - 1 \right), \quad (49)$$

where

$$\delta = \alpha \|A\|_F \|x\|.$$

We wish to apply the above estimate in the regime $t = O(d^2)$. For the sake of specificity, let $t = c_n d^2$. Now, given $\epsilon$, from (49) one can choose $\alpha$ such that

$$\mathbb{P}\left[ \left( \frac{\|\hat{A}_i(x) - A_i(x)\|}{\alpha \|A_i\|_F \|x\|} \le 1 \right) \bigcap A_\epsilon \right] = 0.$$

It suffices to choose $\alpha$ such that when $t = c_n d^2$,

$$2 \left( \frac{\Phi\left( -\frac{d - \delta}{\sqrt{t}} \right)}{\Phi\left( -\frac{d}{\sqrt{t}} \right)} - 1 \right) = \epsilon, \text{ where } \delta = \alpha \|A\|_F \|x\|. \tag{50}$$

Then,

$$\mathbb{P}[A_\epsilon] \le \beta.$$

$\square$

**Remark B.7.** *In the above calculation, the nonlinearity* $\phi$ *can be introduced easily. Clearly, by the compression properties of Algorithm 1, we have that* $\|\hat{A}_i(\phi x) - A_i(\phi x)\| \le \alpha \|A_i\|_F \|\phi x\| \le \alpha\lambda \|A_i\|_F \|x\|$, *where* $\lambda$ *is the Lipschitz constant associated to the nonlinearity* $\phi$. *In particular, if* $\phi$ *is the ReLU, then* $\lambda = 1$. *This gives us that if* $\|\hat{A}_i(x) - A_i(x)\| \le \alpha \|A\|_F \|x\|$,

$$\frac{\|A_i(\phi x) - \hat{A}_i(\phi x)\|}{\|A_i\|_F \|x\|} \le \alpha\lambda \implies \frac{\left| \psi_{E_f}(A_i(x), t) - \psi_{E_f}(\hat{A}_i(x), t) \right|}{\left| \psi_{E_f}(A_i x, t) \right|} \le 2 \left( \frac{\Phi\left( -\frac{d - \delta}{\sqrt{t}} \right)}{\Phi\left( -\frac{d}{\sqrt{t}} \right)} - 1 \right),$$
$$\tag{51}$$

*where*

$$\delta = \alpha\lambda \|A\|_F \|x\|.$$

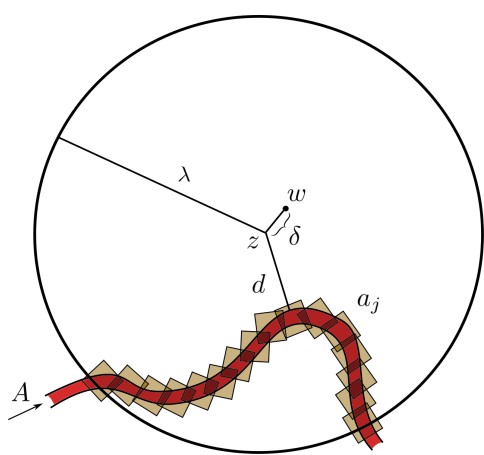

Figure 12: Covering by cuboids of side length $\delta$.

We mention in passing that the above proposition gives a connection between our capacity sensitivity $S(x, A; t)$ and the noise sensitivity $\psi_{\mathcal{N}}$ defined by Arora et al. (2018).

Now consider the case of a curved hypersurface, denoted by $H$ (which is being thought of as the decision boundary $\mathcal{N}_f$), which is "sandwiched" between two hyperplanes $H_1$ and $H_3$. Assume that the hypersurface is at a distance $d$ from the point $z$, and the distance between $H_1$ and $H_3$ is $l$.

**Proposition B.8.** *In the above setting, all the conclusions of Proposition B.6 apply to $H$.*

*Proof.* We have that $\left|\psi_{\mathcal{N}_f}(z, t) - \psi_{\mathcal{N}_f}(w, t)\right|$ is less than or equal to the maximum of the quantities $\left|\Phi\left(-\frac{d}{\sqrt{t}}\right) - \Phi\left(-\frac{d+\delta+l}{\sqrt{t}}\right)\right|$, $\left|\Phi\left(-\frac{d}{\sqrt{t}}\right) - \Phi\left(-\frac{d-\delta+l}{\sqrt{t}}\right)\right|$, $\left|\Phi\left(-\frac{d+l}{\sqrt{t}}\right) - \Phi\left(-\frac{d+\delta}{\sqrt{t}}\right)\right|$, $\left|\Phi\left(-\frac{d+l}{\sqrt{t}}\right) - \Phi\left(-\frac{d-\delta}{\sqrt{t}}\right)\right|$. Let $M(d, t)$ denote this maximum. As argued before, $\psi_{\mathcal{N}_f}(z, t) \geq \Phi\left(-\frac{d+l}{\sqrt{t}}\right)$. That gives,

$$\frac{\left|\psi_{E_f}(z, t) - \psi_{E_f}(w, t)\right|}{\left|\psi_{E_f}(z, t)\right|} \leq \frac{M(d, t)}{\Phi\left(-\frac{d+l}{\sqrt{t}}\right)}.$$

The rest of the argument is similar to the proof of Proposition B.6, and we skip the details. $\qquad\square$

Before moving on to the case of controlled curvature, we need a technical lemma. We state it explicitly because it seems to us that it could have potentially other applications.

**Lemma B.9.** *Let $p \in \mathbb{R}^n$, and consider a cuboid $Q \subset \mathbb{R}^n$ with side lengths $a_1, \cdots, a_n$. Let $q \in Q$ be the unique point which attains $d = \|p - q\| = \mathrm{dist}(p, Q)$. Lastly, assume that the line segment $\overline{pq}$ is perpendicular to the side of $Q$ on which $q$ lies. Then*

$$\psi_Q(p, t) = 2^n \left(\Phi\left(-\frac{a_1}{\sqrt{t}}\right) - \Phi\left(-\frac{a_1 + d}{\sqrt{t}}\right)\right) \prod_{j=2}^{n} \left(\Phi\left(\frac{a_j}{2\sqrt{t}}\right) - \Phi\left(-\frac{a_j}{2\sqrt{t}}\right)\right). \qquad (52)$$

*Proof.* The proof follows easily from the fact that in an $n$-dimensional Brownian motion, all the coordinates execute the standard 1-dimensional Brownian motion independently, and then by applying the reflection principle. The ideas are very similar to the proof of Lemma 3.2 of the main text. $\qquad\square$

As an immediate application of Lemma B.9, we now show that the nice properties of the decision boundaries as mentioned in Propositions B.6 and B.8 above are also shared by hypersurfaces with controlled curvature.

**Proposition B.10.** *Let $H$ be a hypersurface which is diffeomorphic to a hyperplane, of curvature $\kappa$ (in the sense of (26)) satisfying $r \leq \kappa \leq R$. Then the conclusion of Proposition B.6 applies to $H$.*

*Proof.* Let $z$ be a point such that $d := \text{dist}(x, H)$, and $w$ be another point such that $z - w = \delta$. Let $E$ denote the misclassification region defined by $H$.

$$|\psi_E(z,t) - \psi_E(w,t)| \leq \psi_A(z,t),$$

where $A$ denotes the region "sandwiched" between $H$ and $H - \delta$. As before, we will ultimately use $t$ in the regime $O(d^2)$. Now, given $t$, start by considering a ball $B(z, \lambda_t)$, and let $A_{\lambda_t} := A \cap B(z, \lambda_t)$. Here, $\lambda_t$ has been chosen so that $\psi_{A_{\lambda_t}}(z,t)$ comes arbitrarily close to $\psi_A(z,t)$. We will now cover $A_{\lambda_t}$ with $N$ cubes $Q_j, j = 1, \cdots, N$ such that each cube $Q_j$ has sidelengths comparable to $\delta$. Due to the controlled curvature, we know that the cover has controlled multiplicity and

$$N \sim_{r,R,\lambda_t} 1/\delta^{n-1}.$$

Since we know that

$$\psi_{A_{\lambda_t}}(z,t) \leq \sum_{j=1}^{N} \psi_{Q_j}(z,t),$$

it suffices to prove that the RHS above is $O(\delta)$. Via Lemma B.9 above, it suffices to prove the following:

$$\int_{-a}^{a} e^{-x^2} \, dx = O(a).$$

Now, we employ the following known trick:

$$\left( \int_{-a}^{a} e^{-x^2} \right)^n = \int_{-a}^{a} e^{-r^2} r^{n-1} \, dr \, d\omega$$

$$= 2 \int_{0}^{a^2} e^{-\rho} \rho^{n/2-1} \, d\rho$$

$$= 2\gamma(n/2, a^2),$$

where $\gamma(s, x)$ denotes the usual lower incomplete Gamma function. From well-known asymptotics, it is now clear that for small enough $a$, the RHS is $O(a)$. $\qquad \square$

## B.5 Compression parameters: general case

Now we go for the full neural net compression, which is essentially an iterated version of Proposition B.5. Consider a neural net $A$ consisting of $m$ layers, and let $\hat{A}_j$ denote the neural net $A$ whose first $j$ layers have been compressed using the scheme in Algorithm 1 at each level. By way of notation, let $A^j$ denote the $j$th layer of the original neural net (assumed to be of dimension $h_j^1 \times h_j^2$), and $\hat{A}^j$ the $j$th layer of the compressed neural net. Then, we have the following

**Proposition B.11.** *Given $\varepsilon > 0$ and $m$ parameter pairs $(\alpha_j, \beta_j)$, we can find a compression $\hat{A}_m$ with $\sum_{j=1}^{m} O\left(\log(h_j^1 h_j^2)/\beta_j \alpha_j^2\right)$ parameters and associated parameters $\rho_j$ such that*

$$\left| \psi_{E_f}(Ax,t) - \psi_{E_f}(\hat{A}_m x, t) \right| \leq \sum_{j=1}^{m} \rho_j < \varepsilon$$

*with probability at least $\prod_{j=1}^{m} \tau_j$, where*

$$\tau_j = \prod_{i=1}^{j} \left[ (1 - \beta_i) - S(\hat{x}^{j-1}, A_j; t) \right].$$

*Proof.* We see that

$$\left| \psi_{E_f}(Ax,t) - \psi_{E_f}(\hat{A}_m x, t) \right| \leq \left| \psi_{E_f}(Ax,t) - \psi_{E_f}(\hat{A}_1 x, t) \right| + \left| \psi_{E_f}(\hat{A}_1 x, t) - \psi_{E_f}(\hat{A}_2 x, t) \right|$$

$$+ \left| \psi_{E_f}(\hat{A}_2 x, t) - \psi_{E_f}(\hat{A}_3 x, t) \right| + \cdots + \left| \psi_{E_f}(\hat{A}_{m-1} x, t) - \psi_{E_f}(\hat{A}_m x, t) \right|.$$

We will be compressing one individual layer at at time. At the first layer, we start with the entry $x$ taken from the sample set.

Algorithm 1 gives us a compression $\hat{A}^1$ that satisfies, with given $\alpha_1, \beta_1$ that

$$\|A^1 x - \hat{A}^1 x\| \leq \alpha_1 \|A^1\|_F \|x\|$$

with probability at least $1 - \beta_1$. Here the reduced number of parameters of $\hat{A}^1$ is $O\left(\log(h_1^1 h_1^2)/\beta_j \alpha_j^2\right)$. As a result,

$$\left|\psi_{E_f}(Ax, t) - \psi_{E_f}(\hat{A}_1 x, t)\right| \leq \rho_1,$$

where in the general situation (that is, without any additional assumption on the decision boundary $\mathcal{N}_f$), $\rho_1 = \psi_{E_f}(\phi(A_1 x), t)\delta_1$ with probability at least $1 - S(x, A_1; t)/\delta_1 - \beta_1$ (this is via Proposition B.5, via application of Markov's inequality).

Now that the first layer has been compressed, the entry data at the second layer is the vector $\phi \hat{A}^1 x$. Once again, we estimate that with given parameters $\alpha_2, \beta_2$, Algorithm 1 generates a contraction $\hat{A}^2$ at the second layer with satisfies (with probability at least $1 - \beta_2$)

$$\|A^2(\phi \hat{A}^1 x) - \hat{A}^2(\phi \hat{A}^1 x)\| \leq \alpha_2 \|A^2\|_F \|\phi \hat{A}^1 x\|$$
$$\leq \lambda \alpha_2 \|A^2\|_F \|\hat{A}^1 x\| \quad \text{(Lipschitz-ness of the nonlinearity)}$$

So, with probability at least $(1 - \beta_2)(1 - \beta_1)$, we have that

$$\|A^2(\phi \hat{A}^1 x) - \hat{A}^2(\phi \hat{A}^1 x)\| \leq \lambda \alpha_2 \|A^2\|_F \left[\|A^1 x\| + \alpha_1 \|A^1\|_F \|x\|\right]$$
$$\leq \lambda \alpha_2 \|A^2\|_F \left[\|A^1\|_F \|x\| + \alpha_1 \|A^1\|_F \|x\|\right]$$
$$= \lambda \alpha_2 (1 + \alpha_1) \|A^2\|_F \|A^1\|_F \|x\|.$$

We have then

$$\left|\psi_{E_f}(\hat{A}_1 x, t) - \psi_{E_f}(\hat{A}_2 x, t)\right| \leq \rho_2,$$

where in the general situation, $\rho_2 = \psi_{E_f}(\phi(A^2 \hat{x}^1), t)\delta_2$ with probability at least $(1 - \beta_1)(1 - \beta_2) - S(\hat{x}^1, A_2; t)/\delta_2$. Here $\hat{x}^j$ denotes the output at the $j$th layer of the compressed net.

It can be checked via induction that the above process iterated $j$ times gives that

$$\|A^j(\phi(\hat{x}^{j-1})) - \hat{A}^j(\phi(\hat{x}^{j-1}))\| \leq \lambda^{j-1} \alpha_j \prod_{i=1}^{j-1}(1 + \alpha_i) \prod_{i=1}^{j} \|A_i\|_F \|x\|$$

with probability at least $\prod_{i=1}^{j}(1 - \beta_i)$. That implies that

$$\left|\psi_{E_f}(\hat{A}_{j-1} x, t) - \psi_{E_f}(\hat{A}_j x, t)\right| \leq \rho_j,$$

where in the general situation, $\rho_j = \psi_{E_f}(\phi(A_j \hat{x}^{j-1}), t)\delta_j$ with probability at least $\tau_j = \prod_{i=1}^{j}(1 - \beta_i) - S(\hat{x}^{j-1}, A_j; t)/\delta_j$.

Finally, this implies that

$$\left|\psi_{E_f}(Ax, t) - \psi_{E_f}(\hat{A}_m x, t)\right| \leq \sum_{j=1}^{m} \rho_j, \tag{53}$$

with probability at least

$$\prod_{j=1}^{m} \tau_j,$$

and the reduced number of parameters in the compressed net is

$$\sum_{j=1}^{m} O\left(\log(h_j^1 h_j^2)/\beta_j \alpha_j^2\right).$$

$\square$

## B.6 COMPRESSION PARAMETERS: TAME DECISION BOUNDARY

We are left to indicate the proof of the second part of Proposition 5.2 from the main text. This follows in a straightforward way following the proof of Proposition B.11 using the bounds in Propositions B.6, B.8 and B.10 at every step, instead of the bounds in Proposition B.5, as we have done in the above proof.

## B.7 SECOND (ALTERNATIVE) DEFINITION OF CAPACITY SENSITIVITY

As an alternative working definition of noise sensitivity, we define the following:

**Definition 5.**

$$S(x, A; t) := \mathbb{E}_{\gamma \in \mathcal{B}, \eta \sim \mathcal{N}} \left| \frac{\psi_{E_f, \gamma}(\phi(A(x + \|x\| \eta)), t) - \psi_{E_f}(\phi(Ax), t)}{\psi_{E_f}(\phi(Ax), t)} \right|, \tag{54}$$

*where the expectation is over $\eta \in \mathcal{N}$ and all Brownian paths $\gamma$ starting at the point $\phi(A(x + \|x\| \eta))$ and ending inside $E_f(y)$ within time $t$ (the latter sits inside the path space starting at $\phi(A(x + \|x\| \eta))$ and endowed with the Wiener measure). The random variable $\psi_{E_f, \gamma}(\phi(A(x + \|x\| \eta)), t)$ is defined as $1$ if the path $\gamma_l$ strikes $E_f$ within time $t$ and $0$ if it does not.*

From the point of view of ML computation, Definition 5 has a slight advantage over Definition 4. In other words, it is computationally more efficient in view of the following sampling scheme:

**Proposition B.12.** *If $\eta_1, ..., \eta_m$ denote $m$ sampled values of $\eta$ and $\gamma_{j1}, \gamma_{j2}, ..., \gamma_{jk}$ denote $k$ sampled Brownian paths starting at $x + \|x\| \eta_j$, then*

$$\overline{X} = \frac{1}{mk} \sum_{j=1}^{m} \sum_{l=1}^{k} X_{jl},$$

*where*

$$X_{jl} = \left| \frac{\psi_{E_f, \gamma_l}(\phi(A(x + \|x\| \eta_j)), t) - \psi_{E_f}(\phi(Ax), t)}{\psi_{E_f}(\phi(Ax), t)} \right|$$

*approximates $S(x, A; t)$ well with high probability.*

*Proof.* Begin by sampling $m$ values $\eta_1, ..., \eta_m$ of $\eta$ and $k$ Brownian paths $\gamma_{j1}, \gamma_{j2}, ..., \gamma_{jk}$ starting from each such $x + \|x\| \eta_j$. Attached to each such selection is an independent random variable $X_{jl} \psi_{E_f}(\phi(Ax), t)$ which takes values in $[0, 1]$. For each $j, l$, we have that $\mathbb{E}\left(X_{jl} \psi_{E_f}(\phi(Ax), t)\right) = S(x, A; t) \psi_{E_f}(\phi(Ax), t)$. Let $\overline{X}$ denote the mean of all the random variables $X_{jl}$, $j = 1, .., m$, $l = 1, ..., k$. Now, we can bring in Hoeffding's version of the Chernoff concentration bounds, which gives us that

$$\mathbb{P}\left( \left| \overline{X} - S(x, A; t) \right| \geq \frac{\tau}{\psi_{E_f}(\phi(Ax), t)} \right) \leq e^{-2\tau^2 mk}. \tag{55}$$

$\square$

## C APPENDIX C: DATASETS, SAMPLING DETAILS, TRAINING DETAILS AND FURTHER EXPERIMENTS.

### C.1 TECHNICAL SETUP

The experimental section of the work was conducted mainly on a CUDA 10.2 GPU-rack consisting of four NVIDA TITAN V units: this includes the model training as well as Brownian motion sampling and further statistics. The neural network framework of choice was PyTorch 1.5. We provide the training as well as the sampling code for our experiments.

### C.2 DATASETS

We worked with the well-known MNIST and CIFAR-10 datasets. The MNIST is a 784-dimensional dataset that consists of 60000 images of handwritten digits whose dimensions are $(28, 28)$; 50000 images were used for training and 10000 for validation. CIFAR-10 is collection of 60000 32-by-32 color images (i.e. a 3072-dimensional dataset) corresponding to 10 different classes: airplanes, cars, birds, cats, deer, dogs, frogs, horses, ships and trucks; 50000 images were used for training and 10000 for validation.

As pointed out in the main text, adversarially robust decision boundaries exhibit fundamental differences between the MNIST and the CIFAR-10 dataset. MNIST yields particularly simple robust boundaries stemming from it's almost binary nature as elaborated in Schmidt et al. (2017) and confirmed in Ford et al. (2019). CIFAR-10 on the other hand is notoriously vulnerable to attacks, which is reflected in the quantities we measure. For our experiments this means that adversarial/noisy training flattens the surrounding boundary, i.e. saturates the isoperimetric bound, but nevertheless still exhibits spiky structure as will be reflected in the measurements of the isocapacitory bounds. For MNIST on the other hand the approximately binary nature of the examples gives the decision boundary much less 'freedom', resulting in a less distinct quantitative representation.

For some exploratory toy-examples (cf. Fig. 1, Fig. 2, Fig. 3 in the main text) we generated a planar dataset that alternates along a circle of radius $r = 5$: for a given ray through the origin we generate several points on the ray at approximately distance $r$ from the origin and assign them to class 0; then we rotate the ray by a small angle counter-clockwise, sample several points on the rotated ray again at approximately distance $r$ from the origin and this time assign them to class 1. Repeating this process we produce the mentioned 2-class dataset that alternates along the circle of radius $r$ and consists of 1250 points.

### C.3 SAMPLING DETAILS

An evaluation of the isocapacitory saturation $\psi$ is obtained by sampling 10000 Brownian paths with 400 steps. In light of *the curse of dimensionality*, this configuration seems adequate for our purposes: theoretically, by projecting Brownian motion along the normal directions of the decision boundary one sees that estimating hitting probabilities is *essentially* a lower dimensional problem, e.g. 1-dimensional if the decision boundary is a hyperplane; practically, our experiments were numerically stable w.r.t. resampling and sample-batch-size.

Further, for each data point $x$ the relative error volume $\mu(x, r)$ is computed by sampling 10000 points uniformly in $B(x, r)$. To compare with isoperimetric bounds (Subsection 3.2) for each data point $x$ we sample 1000 points, normally distributed $N(x, r/\sqrt{n})$ and concentrated around $x$ in the ball $B(x, r)$, and apply a PGD with 400 steps to obtain distance to the decision boundary $\mathcal{N}$ (a setup similar to Ford et al. (2019)). As above, repetitive runs on average reveal an acceptable numeric stability to the order of $10^{-4}$.

### C.4 DEFENSE TRAINING: FGSM VS PGD

In the present work we are interested in how adversarial/noise defense training are reflected geometrically. To this end we study the application of two defense strategies - FGSM and PGD.

Previous work (Ford et al. (2019)) indicates that FGSM-based training already leads to boundary flattening. However, in general it cannot be guaranteed that the FGSM-based adversarial training will

provide appropriate levels of robustness (against strong adversaries, e.g. iterative attacks) - recently, Wong et al. (2020) has shown that only with some proper designs (e.g. random start) the FGSM-based training will be robust. This indicates that if not taken carefully, FGSM-based and stronger defense trainings (e.g. PGD-based adversarial training in Madry et al. (2018)) can be very different in their resulting geometry of the decision boundary. Therefore, we opt for evaluating FGSM-based as well as the PGD-based defense in an attempt to reveal the relationship between the decision boundaries of a truly robust model and the isocapacitory saturation values. Details are given in Fig. 4 and the accompanying analysis.

## C.5 TRAINING DETAILS

**Training on the CIFAR-10 dataset.**  All training procedures used standard techniques for data augmentation such as flips, horizontal shifts and crops and were normed with respect to data mean and standard deviation. The training of the Wide-ResNets followed the framework provided by Cubuk et al. (2018) with weight decay 5e-4, batch size 128 and a decrease of the initial learning rate of 0.1 by a factor 0.2 at epochs 60, 120 and 160. The ResNets were trained with weight decay 1e-4 respectively and step wise decrease of the learning rate 0.1 by a factor 0.1 at epochs 100 and 150.

**Training on the MNIST dataset.**  We consider two models trained with various data augmentation techniques. We trained a LeNet-5 architecture LeCun et al. (1998) over 50 epochs with a learning rate 1e-3 and weight decay 5e-4, batch size of 64, while optimizing cross entropy loss using root mean square propagation. The same procedure was implemented to train a basic convolutional neural network consisting of four convolutional and two subsequent linear layers. While LeNet-5 also uses convolutional layers, it additionally uses max-pooling after each convolutional layer.

**Training on the planar toy dataset.**  We experimented with several 5-layer MLP models (each layer containing 20, 40, 70 or 100 hidden units) on the mentioned planar dataset concentrated along the circle of radius 5 centered at the origin. Training followed a straightforward ADAM optimization procedure with a learning rate of 1.0e-5 and batch size of 128.

## C.6 DATA MANIPULATIONS DURING TRAINING

To evaluate how various training methods affect the geometric properties of the decision boundary, for all models we conduct three major types of training: training on clean data; on data with a layer of Gaussian perturbations with variance $\sigma^2 = 0.4$; finally, training on data with additional adversarial defense methods, where for each training example we add an adversarially chosen example to the dataset using the fast gradient sign method (FGSM). For LeNet-5 we also considered the effect of adversarial training, where the additional example is the result Brownian of random walk terminated upon collision with the decision boundary. See Fig. C.7 for a visual example of perturbations/attacks with the described methods. The resulting accuracies evaluated on the clean datasets for all trained models are shown in tables 1, 2, 3. As an additional benchmark of the trained models, we evaluated the the robustness of LeNet-5 architectures. Figure C.7 exhibits the resulting for the trained model's accuracies on clean data, PGD attacks with $\epsilon = 0.5$ and $\epsilon = 1.0$, Gaussian perturbations and fog with severity 4 according to the MNIST-C dataset Mu & Gilmer (2019).

## C.7 ISOCAPACITORY AND ISOPERIMETRIC RESULTS

Here we summarize the observations indicated by the obtained geometric data. Besides the results presented in the main text for models Wide-ResNet 28-10 and LeNet-5 (Fig. 4), we also considered geometric properties for said Residual Networks (CIFAR-10) (see Fig. 13) with 32, 44 and 56 layers and a basic Convolutional Neural Network (MNIST) (see Fig. C.7). The results admit to the observations made in the main text.

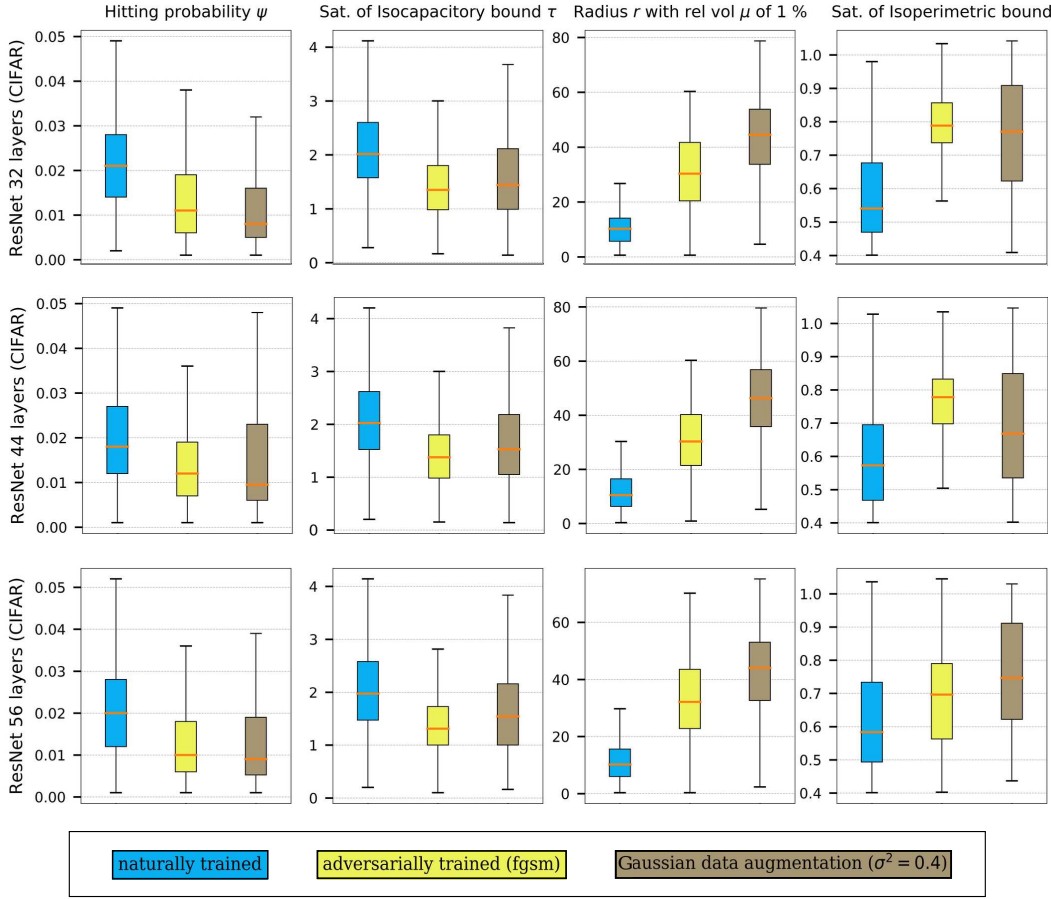

Figure 13: The statistics obtained from the Residual Networks with 32, 44 and 56 layers on the CIFAR10 dataset. For this experiment we considered the Brownian particles with average displacement equal to the radius of sphere with relative volume $\mu = 0.01$, where $\mu$ is defined according to equation (2) in the main text. The considered quantities are *(Left)* the probability of a Brownian particle to collide with the decision boundary, *(Center Left)* the isocapacitory bound, i.e. the ratio of said probability versus relative volume $\mu$, *(Center Right)* the radius of the obtained sphere equal to the RMSD of the particle and *(Right)* the saturation of the isoperimetric bound. We observe consistent behavior of the shown quantities for all three models. The trend of isoperimetric saturation (although, not so concentrated as in the case of WRN and LeNet-5, Fig. 4) as well as the increase of distances $r$ are present. Again the isocapacitory saturation does not appear to follow a distinguished concentration around the case of a flat decision boundary despite the overall increase in flatness: here both noisy and adversarial training seem to deliver a decrease in $\tau$. In fact, the heat imprint of the ordinarily trained model exhibits a "flatter" behaviour in terms of $\tau$.

Table 1: Summary of validation accuracies for Wide-ResNets 28-10 for various training methods on the CIFAR10 data set.

| Architecture | Training Type | Accuracy |
|---|---|---|
| Wide-ResNet 28-10 | naturally trained | 94.64% |
| Wide-ResNet 28-10 | trained on noise ($\sigma^2 = 0.1$) | 91.22 % |
| Wide-ResNet 28-10 | trained on noise ($\sigma^2 = 0.4$) | 86.07 % |
| Wide-ResNet 28-10 | adversarially trained (fgsm) | 87.10 % |
| Wide-ResNet 28-10 | adversarially trained (pgd) | 85.05 % |

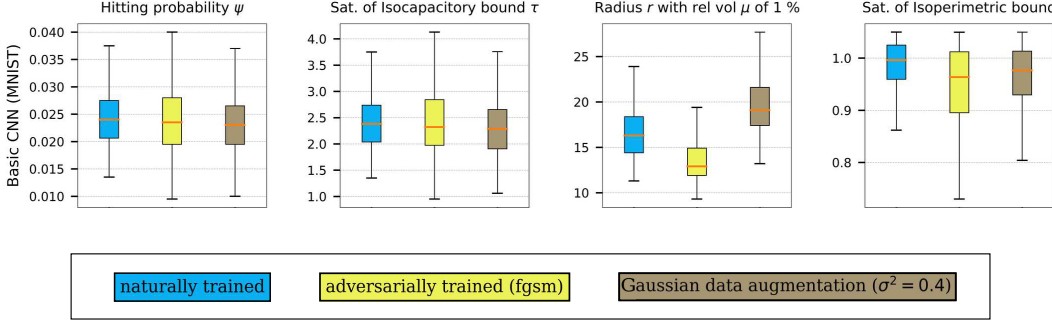

Figure 14: Statistics for a convolutional neural network with four convolutional and two linear layers applied to the MNIST dataset. This particular convolutional model shows that not every architecture/training/dataset instance displays the distinguished trend in increasing the isoperimetric saturation - however, even in this scenario the isoperimetric saturation is quite sharp. Similar to other experiments above, the isocapacitory saturation $\tau$ on the other hand does not concentrate to such an extent.

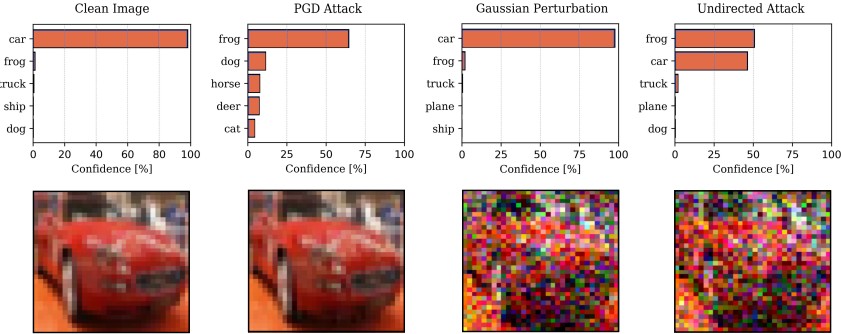

Figure 15: Typical examples of the CIFAR-10 dataset used to train the models. From left to right, the clean image, a PGD adversarial example, a Gaussian perturbation ($\sigma^2 = 0.4$) and the terminal point of a Brownian random walk (undirected attack) immediately after colliding with the decision boundary are shown. The comparison between the PGD adversarial example and the right picture emphasize the degree to which spikes in the decision boundary deviate from the average distance between boundary and clean example.

Table 2: Summary of validation accuracies for the ResNets with 32, 44 and 56 layers for various training methods on the CIFAR10 data set.

| Architecture | Training Type | Accuracy |
|---|---|---|
| Residual Network 32 layers | naturally trained | 91.81% |
| Residual Network 32 layers | adversarially trained (fgsm) | 86.13% |
| Residual Network 32 layers | trained on noise ($\sigma^2 = 0.4$) | 84.36% |
| Residual Network 44 layers | naturally trained | 92.36% |
| Residual Network 44 layers | adversarially trained (fgsm) | 88.20% |
| Residual Network 44 layers | trained on noise ($\sigma^2 = 0.4$) | 84.09% |
| Residual Network 56 layers | naturally trained | 92.77% |
| Residual Network 56 layers | adversarially trained (fgsm) | 87.53% |
| Residual Network 56 layers | trained on noise ($\sigma^2 = 0.4$) | 84.09% |

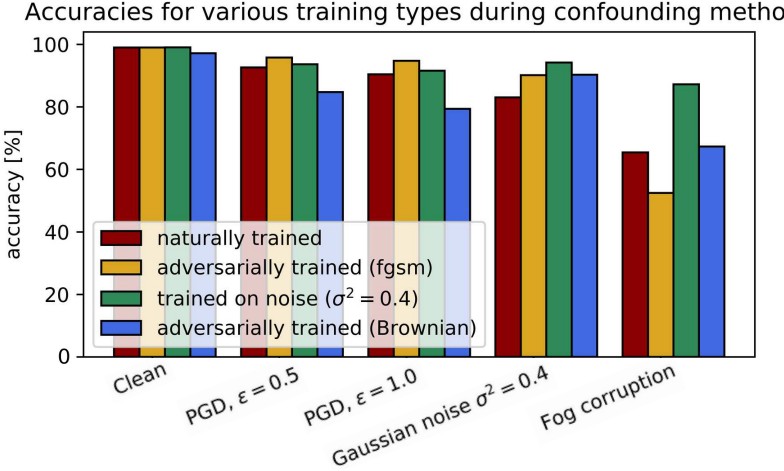

Figure 16: Evaluation of the accuracies of the LeNet-5 (MNIST) models during a range of attacks. While for clean data all models exhibit almost similar accuracy, the adversarially trained models exhibit more robustness during various attacks. For all measures we see the worst performance of the models trained on randomly chosen adversarial examples.

Table 3: Summary of validation accuracies for LeNet-5 and a convolutional neural network with four convolutional and two linear layers for various training methods on the clean MNIST data set.

| Architecture | Training Type | Accuracy |
|---|---|---|
| LeNet-5 | naturally trained | 99.00% |
| LeNet-5 | adversarially trained (fgsm) | 98.99% |
| LeNet-5 | adversarially trained (pgd) | 98.55% |
| LeNet-5 | adversarially trained (Brownian) | 97.17% |
| LeNet-5 | trained on noise ($\sigma^2 = 0.4$) | 99.02% |
| CNN | naturally trained | 98.99% |
| CNN | adversarially trained (fgsm) | 98.65% |
| CNN | trained on noise ($\sigma^2 = 0.4$) | 98.93% |

