# OpenReview forum: "Heating up decision boundaries: isocapacitory saturation, adversarial scenarios and generalization bounds"
_ICLR.cc/2021/Conference — ICLR 2021 Poster_

### Official Review · AnonReviewer4 · 2020-10-29
**Idea interesting but writing can be improved**

**Rating:** 6
**Confidence:** 3

**Review:**

The paper proposes an isocapacitory measure for analysing decision bound, in a way complementing the isoperimetric analysis proposed by Ford et al. 2019. The authors showed that the new measure captures different geometric properties of the decision boundary, potentially useful for adversarial training. The paper also proposed a new generalisation bound, although did not compare with other generalisation bounds.

The paper contains a number of interesting ideas and experiments. In general, the combination of heat diffusion and geometric analysis in the context of machine learning seems an interesting angle worth exploring.

The paper in its current form is a bit difficult to read. The key ideas are not clearly separated from the existing work. Some math notations are not clearly defined. In particular, many definitions rely on globally defined constants such as r and t, which has not been made clear to the reader.

Below are some specific comments:

- Equations 2-5 should be clear that r and t are constants in the definitions or inequalities. The assumption r=sqrt(nt) should be made explicit. This affects the definition of c_n in Equation 4.
-  Equation 3 is incorrect. When mu is less than 0.5, the RHS is negative.

---

> ### Author Response · Authors · 2020-11-11
> **Addressing comments of AnonReviewer4**
>
> Thank you for the helpful and valuable remarks! We attempt to address all the comments below.
>
> "The paper also proposed a new generalisation bound, although did not compare with other generalisation bounds."
>
> We completely agree with the Reviewer that in the present work the generalization bound is mainly of a theoretical nature. Its main purpose is to establish an initial connection between generalization and heat diffusion. Indeed, the further more precise quantitative analysis is a natural next step, which would here nevertheless take the main line of ideas astray: one would require a thorough comparison against a very large body of existing generalization bounds (e.g. PAC-based, compression, pruning, etc) in a spirit similar to (Jiang et al, Fantastic Generalization Measures and Where to Find Them, ICLR 2020).
>
> "The paper in its current form is a bit difficult to read. The key ideas are not clearly separated from the existing work."
>
> We have made several significant updates/improvements to the work (please see the overall comments and revision outline above). In particular, we refer the Reviewer to the newly added discussion in Section 2.
>
> "Some math notations are not clearly defined. In particular, many definitions rely on globally defined constants such as r and t, which has not been made clear to the reader. Equations 2-5 should be clear that r and t are constants in the definitions or inequalities. The assumption $r=\sqrt{nt}$ should be made explicit. This affects the definition of $c_n$ in Equation 4."
>
> We have now defined $\mu(x, r)$ and $\tau(x, r)$ to indicate explicitly the r-dependence. Also, we have mentioned explicitly that $r = \sqrt{nt}$ immediately following Equation 4.
>
> "Equation 3 is incorrect. When $\mu$ is less than 0.5, the RHS is negative."
>
> The Reviewer is completely right, and we have stated the isoperimetric inequality from (Ford et al, 2019) more precisely now.

---

### Official Review · AnonReviewer3 · 2020-10-31
**creative, novel and compelling**

**Rating:** 8
**Confidence:** 3

**Review:**

Title: HEATING UP DECISION BOUNDARIES: ISOCAPACITORY SATURATION, ADVERSARIAL SCENARIOS AND GENERALIZATION BOUNDS

Summary of the paper:

The idea of the paper is to introduce a new view on the geometry of the decision boundary of a classifier. Just as we may speak of the "margin" as in large margin methods and the hinge loss, the paper introduces the idea of a heat diffusion from the decision boundary - the amount of heat diffused to the data points gives a more subtle notion of stability. What a creative idea. Even cooler, the authors show that we may utilise the Feynman-Kac duality to now cast this in terms of the probability of a random walk starting at the data to hit the decision boundary. This is appealing because it differentiates between being near to an e.g. long thin decision boundary which approaches from just one side, to being completely surrounded by the decision boundary - something not accounted for by distance to the boundary.

Most of the paper is a really nice and gentle discussion of this idea. I gained some appreciation for the work even though it is extremely technical under the hood, and this is due to the nice writing in the main paper. To allow this the author's had to push their main result to the end of the paper (proposition 4.1) and leave the details to the appendix, but this is a fair trade-off in my opinion.

It is remarkable that in spite of the technicality involved, the authors manage to obtain proposition 4.1, an impressive first application of the new notions.

The appendices are dense, but still nicely written and enjoyable even for the relatively uninitiated such as myself.

Pros:

This paper stands out as genuinely creative and novel. It is written with an enjoyable style which will surely motivate many theoreticians to delve into the details. I only wish I had the time and talent to do so myself.

While primarily theoretical and highly novel, the authors even include some thought provoking experimental results.

Cons:

One might argue whether the structure of the paper is ideal, but I would counter that the paper is extremely pleasant to read, and that it makes more sense to try to lure the reader into a curious mindset rather than bash them on the head with heavy details from the outset. Nicely done.

Recommendation:

I strongly recommend to accept this paper. Even if the details - which I have not checked - prove to have issues, the novelty of the work makes it a must-have in the portfolio of ideas included in the upcoming ICLR.

---

> ### Author Response · Authors · 2020-11-11
> **Addressing comments of AnonReviewer3**
>
> We are highly grateful to the Reviewer for all the valuable comments, and are really humbled by the kind words.
>
> "One might argue whether the structure of the paper is ideal, but I would counter that the paper is extremely pleasant to read, and that it makes more sense to try to lure the reader into a curious mindset rather than bash them on the head with heavy details from the outset. Nicely done."
>
> We have tried our best to make further updates/improvements in the presentation (please see the overall comments and revision outline above). Hopefully this will further improve the readability of the paper. The Reviewer has also mentioned an illustrative example of differentiating between being near a long thin decision boundary which approaches from just one side, to being completely surrounded by the decision boundary. This seems a very nice and elementary way of  illustrating the idea, and we have taken the liberty of adding a variant of this example to Section 2, point 2. of the text.

---

### Official Review · AnonReviewer1 · 2020-11-02
**empirical results on geometric ratios not conclusive on adversarial trained networks**

**Rating:** 5
**Confidence:** 3

**Review:**

The paper under review introduces a number of geometric measures (isoperimetric, isocapacitory ratios that relate to Brownian motion or heat diffusion probabilities) that are applied to study neural network decision boundaries locally.  Specifically, the studies applying the measures to study adversarially trained NN empirically, and there are generalization and network compression bounds analytically proven that are derived that relate to Brownian motion probabilities.    Empirical observations on LeNet and Wide ResNet trained on MNIST and CIFAR showed adversarially trained or noise trained networks did exhibit curvature of the decision boundary, showing finer structure than previously known.

The paper is clear and the application of isoperimetric and isocapacitory measures appears novel, as well as the empirical finding of curvature (determined through the isocapacitory measure) of adversarially trained NNs provides some new insight into the shape of decision boundaries of NN for robustness.  I didn't find the empirical results very convincing: the isoperimetric / isocapacitory measures do not show a clear distinction for adversarially trained networks, which appears close to the ordinary trained networks.  Its not clear whether the training methods are not sufficient to produce a robust enough NN or the measures introduced do not adequately describe the adversarially trained nets.  All seem to exhibit curvature, but I'd assume the adversarially trained ones exhibit less curvature, which is not the case from the expt.

---

> ### Author Response · Authors · 2020-11-11
> **Addressing comments of AnonReviewer1**
>
> Thank a lot for the insightful comment! We attempt to address this below.
>
> "I didn't find the empirical results very convincing: the isoperimetric / isocapacitory measures do not show a clear distinction for adversarially trained networks, which appears close to the ordinary trained networks. Its not clear whether the training methods are not sufficient to produce a robust enough NN or the measures introduced do not adequately describe the adversarially trained nets. All seem to exhibit curvature, but I'd assume the adversarially trained ones exhibit less curvature, which is not the case from the expt."
>
> Indeed, here, as pointed out in the literature (Ford et al, 2019, Schmidt et al 2018), one should distinguish two different cases - CIFAR10 and MNIST models: in short, CIFAR10 is known to be comparatively vulnerable to adversarial attacks, it has been argued this is due to an insufficiently large training set; meanwhile, MNIST is intrinsically robust against $\ell_\infty$ perturbations as has been observed by Schmidt et al 2018. and confirmed in Ford et al 2019. with the reason being the almost binary nature of the data.
> Now, regarding CIFAR10, there is a clear distinction between isoperimetric and isocapacitory behaviour - please see Figures 4 and 13 for WRN and ResNet model analysis, respectively. The results indicate that adversarial/noise-robustness trainings significantly improve and saturate the isoperimetric bounds (also resonating well with the insights in Ford et al, 2019). This means that curvature is overall reduced. However, on the other hand, our isocapacitory saturation tau is not converging to the one corresponding to the flat case. This is not a contradiction - instead, due to our previous analysis, this is because the isocapacitory saturation is "sharper" and is more sensitive to some fine-scale perturbations or "bubbles" in the decision boundary. This is one of the major geometric insights that one gains therewith - please also see the newly introduced Section 2 for a further outline.
> Regarding MNIST and as mentioned above, models exhibit much less "freedom" to shape the decision boundary through the adversarial training - that is why the saturations in this case are not so clearly articulated. Nevertheless, we believe it is important to provide these MNIST results in order to validate the behaviour of $\tau$ in this more rigid setting.
> Concerning the performed adversarial training, a thorough outline of the training details, model accuracies and corruption stability is presented in Appendix C - in short, we utilize well-known and widely-used training procedures such as FGSM and PGD and their performance appears correct, resonating with the literature.
> Finally, as discussed in the last paragraph of the Analysis Section 4, it is natural to seek further control on $\tau$ leading to saturation - we provided a naive Brownian-based approach that indeed affects $\tau$'s behaviour. However, a tighter saturation control relates with control on a fine-scale that is, in general, a challenging future task.
> We agree that these observations are not sufficiently elaborated in the main and supplementary text and therefore added appropriate remarks in the Analysis Section 4 and the Appendix C.2.

---

### Official Review · AnonReviewer2 · 2020-11-02
**Borderline reject; paper lacks coherence/clarity but might be improved with revisions.**

**Rating:** 7
**Confidence:** 3

**Review:**

The contributions of the paper center on i) the introduction of diffusion-related tools for studying classifier decision boundaries; and ii) using those tools to connect decision boundary geometry, adversarial robustness, and generalization. The paper provides an analysis that provides insight into how curvature and local decision boundary geometry is influenced by adversarial defences, suggests a method for checking adversarial robustness, and then makes statements about model generalization based on geometric properties revealed by monte-carlo simulation of diffusions.


Strengths:

The paper brings together beautiful areas of mathematics, probability, and random processes in an effort to characterize decision boundaries of DNNs and the behavior of DNNs on unseen examples, adversarial or otherwise. Curvature, heat diffusions, diffusion geometry, compression play a role. Feynman-Kac duality is leveraged to pass from the intractable analytical methods in the literature to stochastic simulations that can be undertaken by practitioners with data. The paper appears to make some novel links between generalization and decision boundary diffusion geometry, offers an apparently novel analysis of the impact of common adversarial defenses to Brownian adversaries, and at a minimum offers some new insights into how we might think about, and interpret, complex decision boundaries learned by neural nets or other nonlinear classifiers. The paper also follows up with experiments in the context of real applications and complex models. A series of Appendices provide technical details and experimental protocols.


Weaknesses:

The breadth of topics, the range of tools, steps, and quantities seems to have left the authors without enough space to get it all across. While the Appendices provide valuable details, additional definitions and discussion, the main body of the paper lacks clarity, context, and sufficient organization to allow an average reader to follow, or even appreciate, some of the arguments and key contributions. This is potentially the main weakness of the paper. If this were a longer journal submission, it might even make sense to separate the work into two papers: one exploring adversarial learning, and another exploring generalization.

The introduction and motivation sections of the paper could be improved by explicitly stating at a high level what the paper is contributing, and what insights will come out of the analyses. For example, the abstract states: “This leads to new insights concerning the "flattening-of-boundary" phenomenon.” What insights are in store, specifically? Imagine the paper were to be selected for a popular pod-cast, or a 5-minute lightning oral presentation at a conference: how would you distill it down to the essential contributions, components, and logical steps required to arrive at the key results?

Another concern is that while adversarial defense training by studying hitting times is valuable, it feels slightly misplaced/incomplete because an adversary might (often?) follow geodesics/geometry to very quickly obtain an error sample in cases where diffusion distances are high (e.g. a dumbbell). So while it’s helpful to be able to say that a random walk has a low probability of becoming an error sample from a point x, it doesn’t necessarily give us a guarantee about a “determined” adversary, who isn’t constrained at all to follow random walks (“Brownian attacks”).


Recommendation:
Overall I recommend a borderline reject. The paper has some interesting ideas, and probably novel contributions, but needs revising for clarity and conciseness to be digestible or impactful. The paper needs to also be more specific about how the results relate to, leverage, and complement those in the literature/references. For example, the paper says in a few places that conclusions “agree with” reference [X]. Does this mean they have just rederived the same result, offering no additional insight? What is the significance of the agreement, and what does it provide to the community as a takeaway? It would be helpful to call out very concretely what is being contributed, and how it relates/differs relative to the literature.


Other suggestions:

- Perhaps clarify early on in the paper where the Brownian motion is happening. A reader might wonder: Is it in the ambient space? Is it on a graph? Is it on a manifold parameterized by some kind of intrinsic (or local) coordinates extracted from the model?

- Discuss the significance and interpretation of Lemmas 2.2 and 2.3. Why are they introduced? What are they saying intuitively, and how are they going to support your arguments later? There’s very little text around them (maybe due to a space problem and some ruthless trimming to make it all fit!)

- For the sake of completeness, define explicitly how you intend to handle ties in taking the argmax, and by extension, when defining E(y) (i.e. so that N is a subset of E(y)).

- Section 4 seems to have lost sight of the overall desired result, by omitting how L(g) controls L(f). Apologies if I’ve missed something obvious, but it seems like the strategy to control generalization of f in terms of hitting times is to pass to the compressed version, and control that. So where’s the link between generalization of f and generalization of g? Does Def 1 say it?

- Eq (7) -- \phi_{E} should be introduced to make the definition self contained. Discuss why Def 1 is reasonable and what it means intuitively. The paper says it’s an “initial suggestion”. Why did you suggest this particular defn, over others?

- pg. 4 “isocapacitory” results: the analogy abruptly jumps from heat diffusion to “charge” accumulation. Try to link the two and provide a transition (without the reader having to refer to the lengthy appendices).

- It could be interesting to make a connection to heat diffusion classifiers (e.g. Szlam,
Regularization on Graphs with Function-adapted Diffusion Processes, JMLR 2008), in which heat is instead diffused outward from the labels to classify new points, giving a potentially interesting characterization of the decision boundary (by construction). Maybe such methods can locally approximate more complex learning algorithms thereby providing global insight into their decision boundaries?


UPDATE TO REVIEW FOLLOWING AUTHOR REVISIONS AND COMMENTS
---------------------------------------------------------------------------------------------------
I thank (and commend) the authors for their detailed, point-by-point responses.
The authors have made a good effort in their revisions to improve and clarify the exposition, and rectify the other comments made by the reviewers. The paper could still benefit from a deeper rewrite -- there's just so much that can be packed into a conference paper with limited real estate, and the authors are seeking to make several contributions under the umbrella of one submission (as the title suggests). So clarity suffers, and impact will suffer as a result. But, in my mind that shouldn't necessarily be a show stopper at this stage, in light of the revisions. I am therefore upgrading my recommendation.

---

> ### Author Response · Authors · 2020-11-11
> **Addressing comments of AnonReviewer2**
>
>
> We deeply thank the Reviewer for the helpful comments and insights!
>
> 1."The breadth of topics,... enough space to get it all across."
> Please, refer to the general response message above. Making use of the additional 9th page for the main text, we add a new Section 2 where we hope to have delivered a clean outline of our contributions, context and paper roadmap. Also, we made many further small improvements towards readability and text-flow.
>
> 2."For example, the abstract states:...What insights are in store, specifically?"
> Here, the intended geometric message was that "adv. training forces boundaries to become flatter, but the heat traits (to certain surprise) do not correspond to the flat case and this geometrically translates to finer-scale non-linearities in the decision boundary". The sentence in the Abstract is reformulated accordingly and further elaboration of these insights is provided in the new Sections 2 and 1.
>
> 3."So while it’s helpful ... guarantee about a “determined” adversary".
> Indeed, providing robustness algorithms or sophisticated adv. attacks is not a primary goal of the present work. What the "BM attacks" interestingly show is that basic adv. defenses are not necessarily able to impact their behaviour.
>
> 4."For example, the paper says in a few places...offering no additional insight?"
> The "agree with" statements act mainly as sanity checks that the benchmarks (e.g. isoperimetric saturation) are behaving in accordance with previous results. Thus, we have a fair environment where the previous estimates are acting as outlined in the literature and we can compare them against our novel heat metrics, analyzing the utility of our tools. A description of contributions and relations to literature is now provided in the revision. Please, see the new context discussion in Sec. 2.
>
> 5.Updates for each of the additional suggestions:
>
> -Clarification about the BM ambient space is added in the abstract and introduction.
>
> -Concerning Lemmas 2.2 and 2.3 (now 3.2 and 3.3), additional motivation has been given in the new Contributions Section 2 (in the road map) as well as in Section 3.4.
>
> -We have inserted a comment on handling the ties in argmax making N a subset of E(y).
>
> -Concerning L(g) vs L(f) bounds, indeed, one gets a bound on the ability of the compressed model g to generalize. This is the essential statement of the recent compression-based methods (e.g. Arora et al, 2018). The idea is that g is considered almost equivalent to the full model f (in an appropriate sense) and hence gen. bounds on g are quite useful. Moreover, as a second step, one is also able to obtain bounds on the full model f in terms of g via "peeling" (Suzuki et al, 2020). However, in the present work we choose to leave out the detailed discussion of this second step. In Sec. 5, we emphasize this step-wise process before the propositions. At the end of Sec. 5 we add a paragraph on obtaining bounds for the initial model f. We moreover update the Introduction Section 1 and the background on gen. bounds accordingly.
>
> -Concerning Eq. (7) and Def. 1, we have now defined the notation $\psi_E$ in general in Eq. (1). Def. 1 actually seems to be the most automatic analogue of Def. 1 of (Arora et al, 2018) in our setting. Instead of taking a pointwise difference, we aim to capture the idea that the decision boundary of a good compression should be "close enough" to the decision boundary of the original classifier. In our context, this implies that their "heat signatures" at the sample points should be close enough at all time scales. Def. 3 in the Appendix is a close variant of this definition, leading to a closely related result, Prop. B.2. We have now added a couple of sentences explaining this above Def. 3. We chose to feature Def. 1 in the main text as it seemed slightly more intuitive to us than Def. 3.
>
> -Concerning charge/capacity, we agree that the jump seemed abrupt. We have now attempted to provide a smooth transition.
>
> -Finally, we thank the Reviewer for making us acquainted with (Szlam et al 2008). Their diffusion oriented method of assigning labels from known data points to the unknown might be connected to our ideas. For example, consider the case of binary data with a completely trained classifier, and every data point is assigned a label as  "positive" or "negative". Suppose also that we have the idealized situation that data points (of either type) are uniformly distributed around and near the decision boundary. Now, let us start a diffusion process from all the data points, as in (Szlam et al, 2008), and try to reclassify the data according to their method. Then the data points which change their classification in very short time are the ones located near narrow "spikes" in the decision boundary. This idea could be useful in determining spikyness in the decision boundary and locating adv. examples. We have included Szlam et al, 2008 in the Subsection "Additional related work" on page 2.

---

### Author Response · Authors · 2020-11-11
**Thank you for the valuable comments; Revision posted.**

Dear Reviewers,

We are most grateful for all of your valuable and helpful comments and sincerely appreciate the invested time in revising our work - thank you! We took seriously every single raised comment/remark and attempted to address it accordingly.
Overall, it seems to us that the major remark revolves around writing and presentation. Indeed, we perfectly agree that several (fairly extensive) tool sets are interlaced within the present work and the task to communicate and "abstractize" the results/contributions/techniques is of utmost importance. While it might be a challenge to digest some of the individual technical details, we however believe that the main novel contributions and results are quite straightforward and easy to get across.
To this end, we uploaded a new revised version where we focus on precisely this very goal and we sincerely hope the main text is now much improved in terms of context, methods and contributions. Of course, we are glad to utilize the rebuttal period as much as possible and incorporate any further suggestions and changes that would appear in the follow-up discussions.
In short, we describe the current updates as follows:

(1) We utilize the additional 9th page and provide a new Section 2 "Contributions, Context and Outline of Work". We give a list of our essentially novel contributions (also within the context of related work and literature). Furthermore, we outline a "road map" of the paper where the individual steps in the discussion are synthesized and highlighted. We hope this new Section provides not only a clean lucid description of contributions and techniques, but also augments the reader's intuition making the exposition of the main text much more natural and easy to follow.

(2) Many smaller improvements overall the main text that increase the flow of the discussion and make the text more coherent, context-motivated and self-contained.

(3) We addressed each individual comment from the reports (please see the corresponding detailed outline).

We are very much looking forward to your further comments!
Again, thank you for the time to consider the present paper.

With best regards,
the Authors

---

### Author Response · Authors · 2020-11-19
**Thank you for the valuable comments; Revision further updated.**

Dear Reviewers,

We thank you once again for all your support so far.  In addition to our revision last week, we have further taken the liberty to update our paper slightly. There are no significant additions in comparison to the previous revision last week: in the main text, some sentences have been reworded and some explanations have been optimized. In the Appendix, we have included some additional lines of explanation, e.g., Wiener measures, Hoeffding's inequality, hitting probability scaling and such like, resulting in the addition of a page or so worth of content. Our hope is that the updated manuscript will read better and will be more self-contained.

We thank you for your time, and look forward to hearing from you.

Best regards,
The Authors

---

### Author Response · Authors · 2021-01-16
**Thank you for all your support**

Dear Reviewers and Chairs,

We are deeply thankful to you for all your encouragement and supportive comments. We are delighted to be a part of ICLR 2021 and look forward to a wonderful learning experience.
If you have any further comments regarding the exposition, please let us know and we will try to address them in the camera-ready version accordingly.

Best regards,
The Authors

---

### Decision · Program_Chairs · 2021-01-07
**Final Decision**

**Decision:**

Accept (Poster)

**Comment:**

Four reviewers have reviewed this paper and after rebuttal, they were overall positive about the proposed idea. We congratulate authors on the paper.